# An image-based data-driven analysis of cellular architecture in a developing tissue

Jonas Hartmann[1]*, Mie Wong[2], Elisa Gallo[1,2,3], Darren Gilmour[2]*

[1]Cell Biology and Biophysics Unit, European Molecular Biology Laboratory (EMBL), Heidelberg, Germany; [2]Institute of Molecular Life Sciences, University of Zurich (UZH), Zurich, Switzerland; [3]Collaboration for joint PhD degree between EMBL and Heidelberg University, Faculty of Biosciences, Heidelberg, Germany

**Abstract** Quantitative microscopy is becoming increasingly crucial in efforts to disentangle the complexity of organogenesis, yet adoption of the potent new toolbox provided by modern data science has been slow, primarily because it is often not directly applicable to developmental imaging data. We tackle this issue with a newly developed algorithm that uses point cloud-based morphometry to unpack the rich information encoded in 3D image data into a straightforward numerical representation. This enabled us to employ data science tools, including machine learning, to analyze and integrate cell morphology, intracellular organization, gene expression and annotated contextual knowledge. We apply these techniques to construct and explore a quantitative atlas of cellular architecture for the zebrafish posterior lateral line primordium, an experimentally tractable model of complex self-organized organogenesis. In doing so, we are able to retrieve both previously established and novel biologically relevant patterns, demonstrating the potential of our data-driven approach.

*For correspondence:
jonas.m.hartmann@protonmail.
com (JH);
darren.gilmour@imls.uzh.ch (DG)

**Competing interests:** The authors declare that no competing interests exist.

## Introduction

Organogenesis proceeds as a complex multi-scale process. Cells utilize a wide range of molecular machinery in a coordinated fashion to give rise to tissue-scale collective behavior, which in turn feeds back on individual cells' architectural and transcriptional dynamics (*Chan et al., 2017*). Uncovering the principles that govern these systems is a long-standing but elusive goal of developmental biology, in part because it is often challenging if not impossible to reduce such complex phenomena to the action of single genes or simple mechanisms (*Bizzarri et al., 2013*). Thus, there is a persistent need for new techniques that enable integrative analysis of developmental systems.

In recent years, *data science* has arisen as a new interdisciplinary paradigm that combines statistics, computer science and machine learning with the aim of generating knowledge in a data-driven rather than hypothesis-driven fashion (*Dhar, 2013*; *Blei and Smyth, 2017*; *Baker et al., 2018*). Data science thus provides tools to computationally query datasets for patterns that explain the data in an open and unbiased way, not just to test whether the data fit a preformed hypothesis. The application of such data-driven approaches to biology promises a new way of extracting relevant information from large and complicated datasets describing complex biological systems. It thus complements the increasingly rapid pace at which biological data are being generated. However, whilst this promise is already being realized to great effect in some fields, for instance in high-throughput cell biology (*Roukos and Misteli, 2014*; *Gut et al., 2018*; *Chessel and Carazo Salas, 2019*) and in (multi-)omics analysis (*Libbrecht and Noble, 2015*; *Angerer et al., 2017*; *Huang et al., 2017*; *Ching et al., 2018*), developmental biology has seen little adoption of data science techniques to date.

This is primarily because the field's main source of data, in vivo microscopy, does not readily lend itself to the production of 'big data', upon which much of the recent progress in data science is

founded. Although imaging datasets of in vivo biological systems are often large in terms of computer memory, they generally do not benefit from the defining property that makes 'big data' so useful, namely very large sample numbers on the order of thousands or more, which is not easily achievable in most embryonic model systems. In addition, the high degree of sample to sample variance complicates the use of registration techniques to generate averaged 'reference' embryo datasets. Furthermore, only a handful of components can be labeled and observed simultaneously by current fluorescence microscopy methods, which constrains the range of possible biological relationships that could be discovered by applying data science. Despite these limitations, imaging data have the unique advantage that they contain information on the spatial localization of measured components and thus indirectly encode rich higher-order information such as patterns, textures, shapes, locations, and neighborhoods. Moreover, they allow the dynamics of such spatial features to be followed at high temporal resolution. In short, quantitative imaging generates 'rich data' rather than 'big data'.

Progress towards employing the power of data science for the imaging-based study of development faces three challenges: (1) unpacking the rich spatial information encoded in images into a format that is accessible for data science techniques (*data extraction*), (2) integrating data across multiple experiments to overcome the limited number of simultaneous measurements (*data integration*), and (3) analyzing and visualizing the resulting multi-modal dataset to enable the discovery of biologically meaningful patterns (*data interpretation*).

Here, we address each of these challenges in the context of a comprehensive data-driven analysis of cellular architecture in an experimental model tissue, the zebrafish posterior Lateral Line Primordium (pLLP), which migrates along the flank of the developing zebrafish embryo, periodically assembling and depositing rosette-shaped clusters of cells that cease migration and differentiate to form sensory organs (*Haas and Gilmour, 2006*; *Ghysen and Dambly-Chaudière, 2007*). The pLLP is patterned into a leader zone of cells that are highly polarized in the direction of migration and a follower zone where rosettes are being assembled through apical constriction (see Figure 2A; *Nechiporuk and Raible, 2008*; *Lecaudey et al., 2008*). This tight integration of collective migration, patterning and morphogenesis gives rise to an intricate tissue architecture that, although easy to image, is challenging to study at the single-cell level as its cells exhibit a wide variety of shapes and behaviors in addition to being comparably small and densely packed (*Galanternik et al., 2016*; *Nogare et al., 2017*).

To bring data science to this problem, we engineered a multi-step data extraction and analysis pipeline (*Figure 1*). Starting from high-resolution in vivo microscopy data, our pipeline first performs 3D single-cell segmentation and then utilizes a newly developed point cloud-based algorithm to extract the information contained in the volumetric data of each cell into a simple set of numbers, so-called *features*. This enabled us to perform a quantitative and unbiased analysis of cell shape across the pLLP. Next, we sought to incorporate internal cellular structures such as organelles into our analysis in order to investigate the relationship between cell morphology and intracellular organization. We achieved this by co-opting machine learning techniques to integrate data across experiments with different markers into a multi-modal atlas. Finally, we built on this approach to map contextual knowledge onto the atlas as a means of facilitating data interpretation through context-guided visualization. We show that this data science-inspired analysis retrieves both known and novel patterns of how cells are organized within the pLLP, demonstrating how a data-driven approach can lay a quantitative foundation for the systems-level study of organogenesis.

## Results

### High-Resolution live imaging and 3D Single-Cell segmentation of the zebrafish Posterior Lateral Line Primordium

The application of data-driven approaches to relevant problems in developmental biology depends on high quality input datasets of the tissue of interest. A data-driven analysis of cellular architecture thus requires high-resolution imaging followed by automated segmentation of individual cells. As developing embryos are large 3D specimens, it is beneficial to use a microscopy method that allows optical sectioning of defined axial z-planes. Here, we employed AiryScan FAST mode confocal microscopy (*Huff, 2016*) to achieve high signal-to-noise ratios and high axial resolution – both of

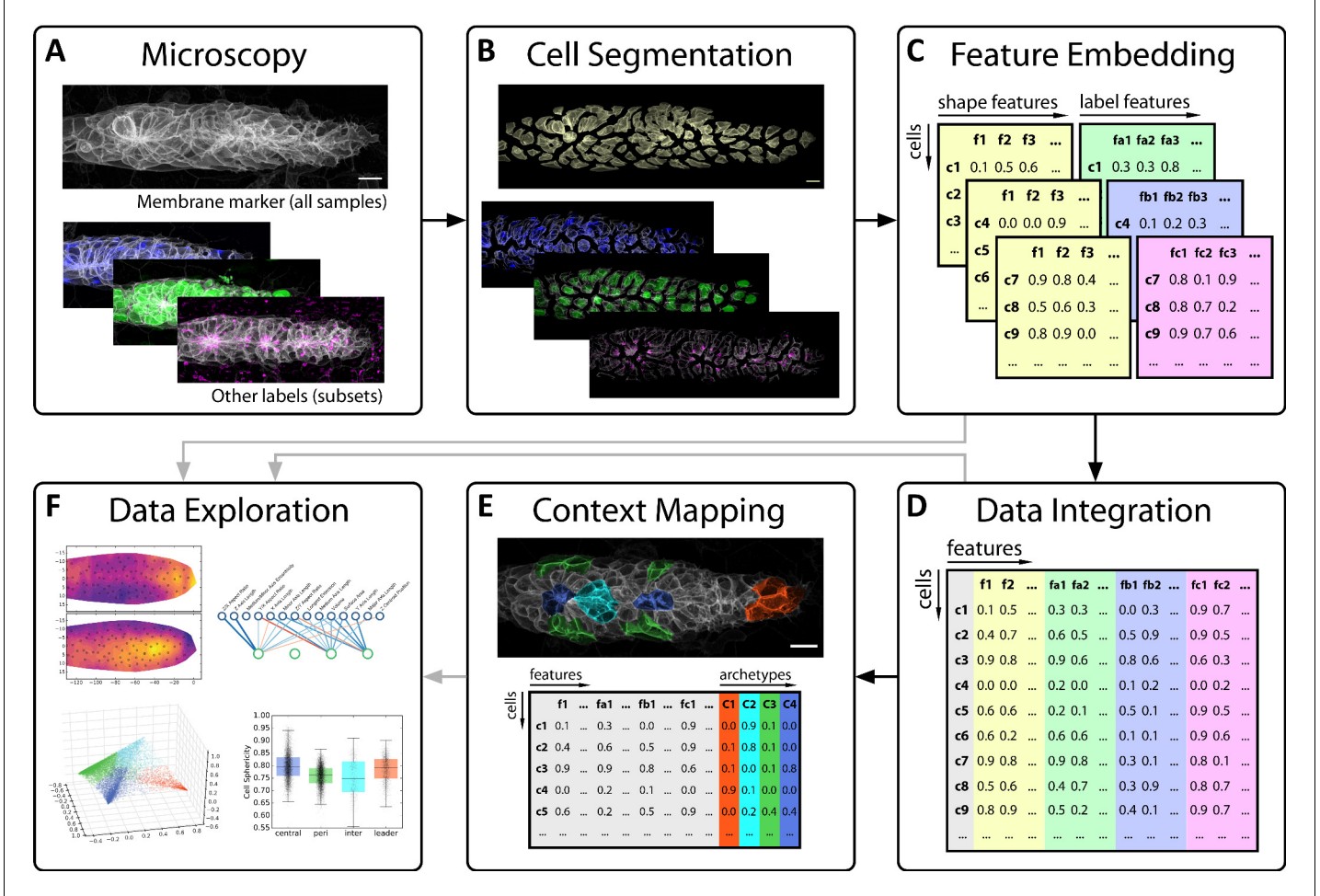

**Figure 1.** Overview of key steps in our data-driven analysis workflow. (A) Image data of the tissue of interest are acquired using 3D confocal fluorescence microscopy. Each sample is labeled with a membrane marker to delineate cell boundaries (top) and samples can additionally be labeled with various other markers of interest (bottom, colored). (B) Using an automated image analysis pipeline, single cells are automatically segmented based on the membrane marker to prepare them for analysis, illustrated here by shifting them apart. (C) Next, data extraction takes place to arrive at numerical features representing the cell shapes (yellow) and the various fluorescent protein distributions of additional markers (other colors). (D) Such well-structured data simplify the application of machine learning techniques for data integration, which here is performed based on cell shape as a common reference measurement. (E) A similar strategy can be used to map manually annotated contextual knowledge (top) into the dataset (bottom), in this case specific cell archetypes chosen based on prior knowledge of the tissue's biology. (F) Finally, all of the resulting data are explored and interpreted through various visualizations and statistics.

which greatly facilitate 3D segmentation – whilst maintaining a sufficiently high acquisition speed for live imaging of a migrating tissue (~20 s/channel). In this way, we acquired a large set of single and dual channel volumes of wild-type primordia during migration, staged such that the pLLP is located above the posterior half of the embryo's yolk extension (32-36hpf, N = 173 samples in total). All of our samples expressed the bright cell membrane label *cldnb:lyn-EGFP* (*Haas and Gilmour, 2006*; *Figure 2A*, *Figure 2—video 1*) and many additionally expressed a red or far-red label for an internal cellular structure (see table in *Supplementary file 1* for a complete overview).

To segment cells, we combined commonplace image processing algorithms into a specialized automated pipeline that uses labeled membranes as its sole input (see materials and methods for details). We found that our pipeline reliably produces high-quality segmentations (*Figure 2B*, *Figure 2—video 2*) and that erroneously split or fused cells are rare. To further ensure consistent segmentation quality, each segmented stack was manually double-checked and rare cases of stacks where more than approx. 10% of cells had been either missed, split or fused were excluded (8 of 173; 4.62%). An expanded visualization of a representative segmentation (*Figure 2C*, *Figure 2—*

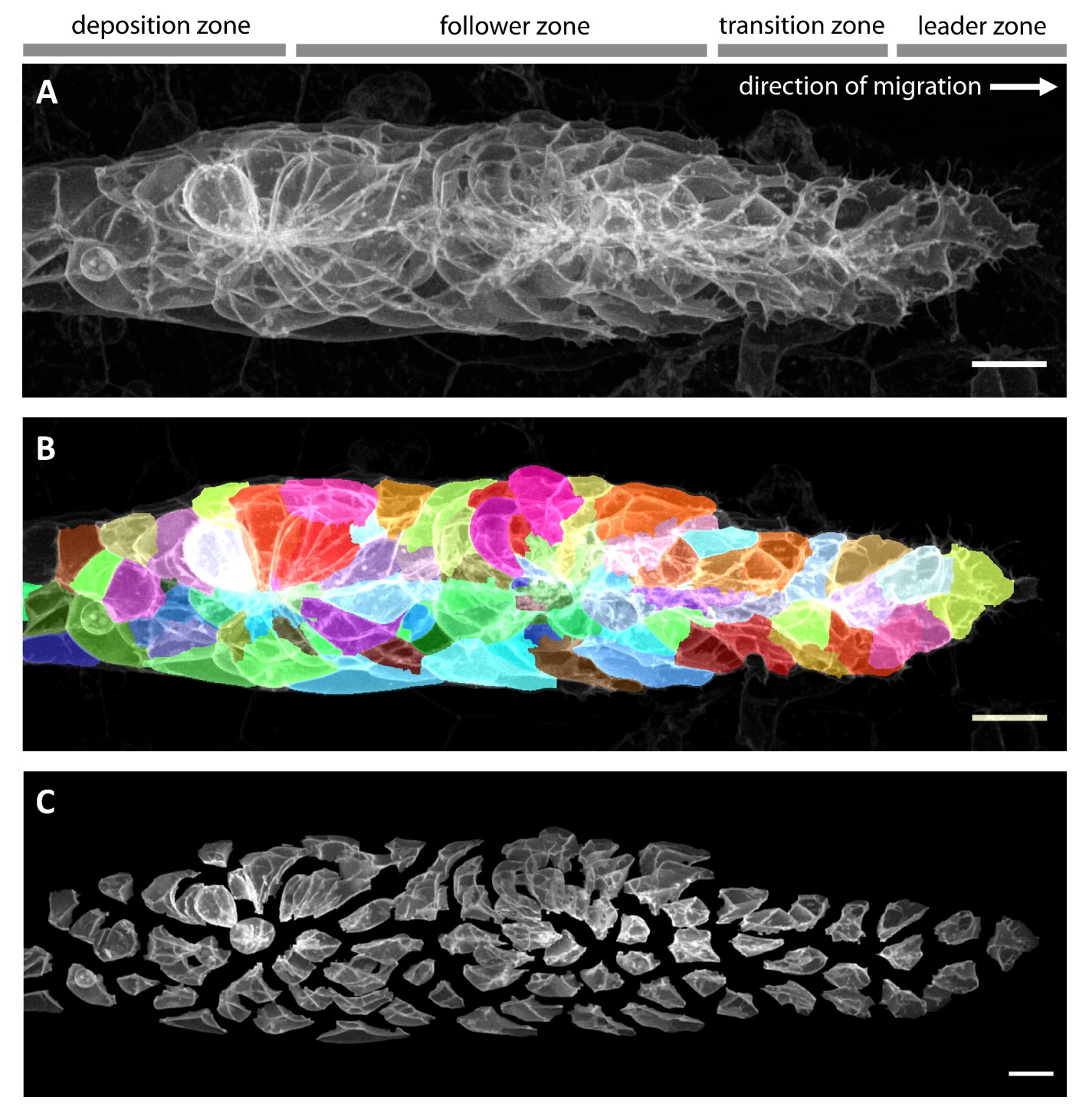

**Figure 2.** Imaging and automated 3D single-cell segmentation of the pLLP. (**A**) Maximum z-projection of a deconvolved 3D volume of the pLLP acquired using the LSM880 AiryScan FAST mode. (**B**) The same primordium shown with a semi-transparent color overlay of the corresponding single-cell segmentation. (**C**) Expanded view of the same primordium; individual segmented cells have been shifted apart without being rescaled or deformed, revealing their individual shapes within the collective. Note that the segmentation faithfully recapitulates the diversity of cell shapes within the pLLP, with the exception of fine protrusions. Since the protrusions of follower cells are often impossible to detect against the membranes of the cells ahead of them, we decided not to include fine protrusions in our analysis. All scale bars: 10 μm.

The online version of this article includes the following video(s) for figure 2:

**Figure 2—video 1.** Dynamic View 3D pLLP Stack.

*Figure 2 continued on next page*

*Figure 2 continued*

**Figure 2—video 2.** Dynamic View of 3D pLLP Segmentation.

**Figure 2—video 3.** Dynamic View of 3D Expanded Segmentation.

*video 3*) allows the diversity of cell shapes in the lateral line and their relationship with overall tissue architecture to be appreciated in a qualitative fashion.

Overall, we collected n = 15,347 segmented cells from N = 165 wild-type primordia as the basis for our quantitative analysis. Unless otherwise specified, the results presented in this study are based on this dataset.

## Point Cloud-Based morphometry for unbiased quantification of cellular architecture

Most data science techniques require input data in a specific form, namely a *feature space*, which consists of a vector of numerical features for each sample and hence takes the shape of a 2D samples-by-features array (see *Figure 1C*). Even when segmented, confocal volumes of cells do not conform to this standard. Thus, numerical features must be extracted from volumetric image data. This can be achieved by simply measuring specific aspects of each cell such as volume or eccentricity (*feature engineering*) but for exploratory analysis it is preferable to derive an unbiased encoding of 3D image information into a 1D feature vector (*feature embedding*). This non-trivial task has previously been tackled in a number of ways (*Pincus and Theriot, 2007*; *Peng and Murphy, 2011*; *Rajaram et al., 2012*; *Tweedy et al., 2013*; *Kalinin et al., 2018*), however no readily applicable solution for feature embedding of both cell morphology and subcellular protein distributions from segmented 3D images has been described to our knowledge.

We therefore developed a novel method that allows feature embedding of arbitrary fluorescence intensity distributions. As a starting point, we took a standard workflow from the field of geometric morphometrics, which consists of four steps (*Adams et al., 2013*; *Figure 3A*):

(1) conversion of image data – which is dense and regularly spaced in voxels – into a sparse point cloud of landmarks to facilitate computational transformations, (2) alignment of point clouds by registration to remove translational and rotational variance and hence retrieve pure shape information, (3) re-representation of landmark coordinates based on their deviation from a consensus reference common to all samples in order to arrive at features that are comparable across samples, and (4) dimensionality reduction by Principal Component Analysis (PCA) to find the most relevant features within the data. To adapt this classical workflow to make it applicable to 3D images of cells (*Figure 3B*, *Figure 3—figure supplement 1*), it was necessary for us to solve three key problems.

First, classical workflows extract point clouds based on specific landmarks that are consistent from sample to sample, such as the nose and eyes in facial shape analysis. As a heterogeneous population of cells does not display such common landmarks, we instead implemented a sampling strategy termed *Intensity-Biased Stochastic Landmark Assignment* (ISLA) inspired by *Chan et al., 2018*. ISLA automatically generates a point cloud that is a sparse encoding of the original image, with the local density of points representing local fluorescence intensity (*Figure 3C*). ISLA achieves this by treating normalized voxel intensity distributions as multinomial probability distributions and sampling them to obtain a predefined number of landmark coordinates. By adjusting the number of landmarks being sampled, the trade-off between precision and computational performance can be controlled.

Second, a solution was needed to prevent differences in cell location, size and rotation from obscuring shape information. We first removed translational variance by shifting all cells such that their centroids are placed at the origin coordinates (0,0) and then removed size variation by rescaling point clouds such that cell volumes are normalized. The removal of rotational variance, however, is usually achieved by spatial registration of cells (*Pincus and Theriot, 2007*), which is an ill-defined problem for a highly heterogeneous cell population. Therefore, we instead chose to re-represent the 3D point cloud of each cell in terms of the points' pairwise distances from each other, a

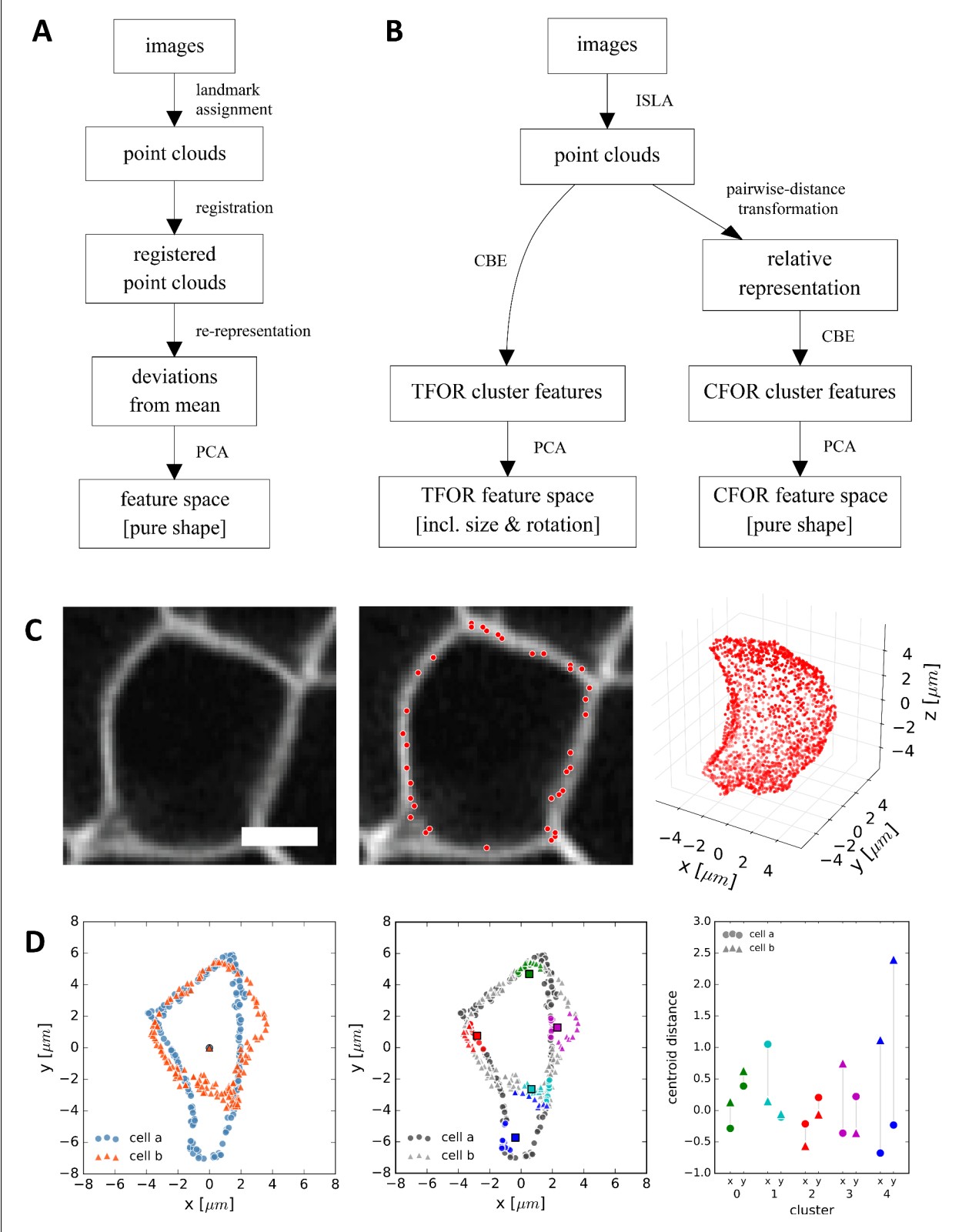

**Figure 3.** CBE and ISLA for Point Cloud-Based Cell Morphometry. (**A**) A classical workflow in landmark-based geometric morphometrics. (**B**) Adapted workflow for morphometrics of arbitrary fluorescence intensity distributions. See *Figure 3—figure supplement 1* for a more detailed version. (**C**) Illustration of ISLA, our algorithm for conversion of voxel-based 3D images to representative point clouds. Shown are a slice of an input image (left), here a membrane-labeled cell in the pLLP (scale bar: 2 μm), the landmarks sampled from this image (middle), here oversampled compared to the
*Figure 3 continued on next page*

*Figure 3 continued*

standard pipeline for illustration purposes, and the resulting 3D point cloud (right). (**D**) Illustration of CBE, our algorithm for embedding point clouds into a feature space. In this 2D mock example, two cells are being embedded based on point clouds of their outlines (left). CBE proceeds by performing clustering on both clouds combined (middle) and then extracting the distances along each axis from each cluster center to the centroid of its ten nearest neighbors (right). Note that the most distinguishing morphological feature of the two example cells, namely the outcropping of cell *a* at the bottom, is reflected in a large difference in the corresponding cluster's distance values (cluster 4, blue).

The online version of this article includes the following figure supplement(s) for figure 3:

**Figure supplement 1.** Flowcharts Illustrating the ISLA and CBE Algorithms.

**Figure supplement 2.** Evaluation of the Expressiveness of CBE Embeddings With and Without Cell Frame of Reference (CFOR) Normalization.

representation that is intrinsically invariant to translation, rotation and handedness but retains all other information on the cloud's internal structure. We use the term *Cell Frame Of Reference* (CFOR) to refer to feature spaces where size and rotation have been normalized in this fashion, so only pure shape information is retained.

While it is beneficial to remove size and rotation when analyzing cell shape, these properties are themselves important for the organization of multicellular systems. For example, dedicated planar cell polarity pathways have evolved to control the rotational symmetry of cells within many tissues, including the mechanosensory hair cell organs deposited by the lateral line primordium (*Mirkovic et al., 2012*; *Jacobo et al., 2019*). Thus, in order to include this level of organization in our analysis, we also pursued a second approach wherein we register entire tissues (rather than individual cells) prior to feature embedding. In this case, rotational variance of individual cells is not removed but instead becomes biologically meaningful as it now reflects the actual orientation of cells within the tissue. Feature spaces resulting from this approach are referred to as *Tissue Frame Of Reference* (TFOR) and retain both cell size and relevant rotational information (*Figure 3B*).

Third and finally, given that point clouds extracted by ISLA are not matched across multiple cells, the classical method for defining a consensus reference across samples, which is to use the average position of matched landmarks (e.g. the average position of all nose landmarks in facial analysis), is not applicable here. To address this, we developed *Cluster-Based Embedding* (CBE) [inspired by *Qiu et al., 2011*], which works by first determining which locations in space contain relevant information across all samples and then measuring each individual sample's structure at these locations. More specifically, CBE proceeds by first performing k-means clustering on an overlay of a representative subset of all cells' point clouds. Using the resulting cluster centers as consensus reference points, CBE then re-represents each individual cell's point cloud in terms of a simple proximity measure relative to said cluster centers (*Figure 3D*). We validated CBE using a synthetically generated point cloud dataset, finding that it outperforms an alternative embedding strategy based on point cloud moments and that normalization of size and rotation (i.e. the cell frame of reference, CFOR) is beneficial in the detection of other shape features (*Figure 3—figure supplement 2*). For further details on feature embedding with ISLA and CBE as well as our evaluation using synthetic point cloud data see the materials and methods section.

In summary, Cluster-Based Embedding (CBE) of point clouds obtained from 3D images by Intensity-biased Stochastic Landmark Assignment (ISLA) is an expressive and versatile embedding strategy for transforming arbitrary 3D fluorescence distributions into 1D feature vectors. It thus provides a solution to one of the key challenges facing data-driven analysis of tissue development by unpacking the rich information encoded in image stacks into a format that is accessible for further quantitative analysis.

## The cellular shape space of a model tissue: The lateral line primordium

Cell shape links cellular mechanics with tissue architecture and is thus key to understanding morphogenesis in model tissues like the developing lateral line primordium. To gain a data-driven view of cell shape in this system, we applied the ISLA-CBE workflow to the cell boundaries of our segmented dataset, deriving an embedded feature space representing cell morphology across the tissue, which we termed the pLLP's *cellular shape space*. We found that the resulting Principal Components (PCs) describe meaningful shape variation (*Figure 4A–B*) and that a small number of PCs is sufficient to capture most of the shape heterogeneity across the tissue (*Figure 4C*). Interestingly, the cells of the

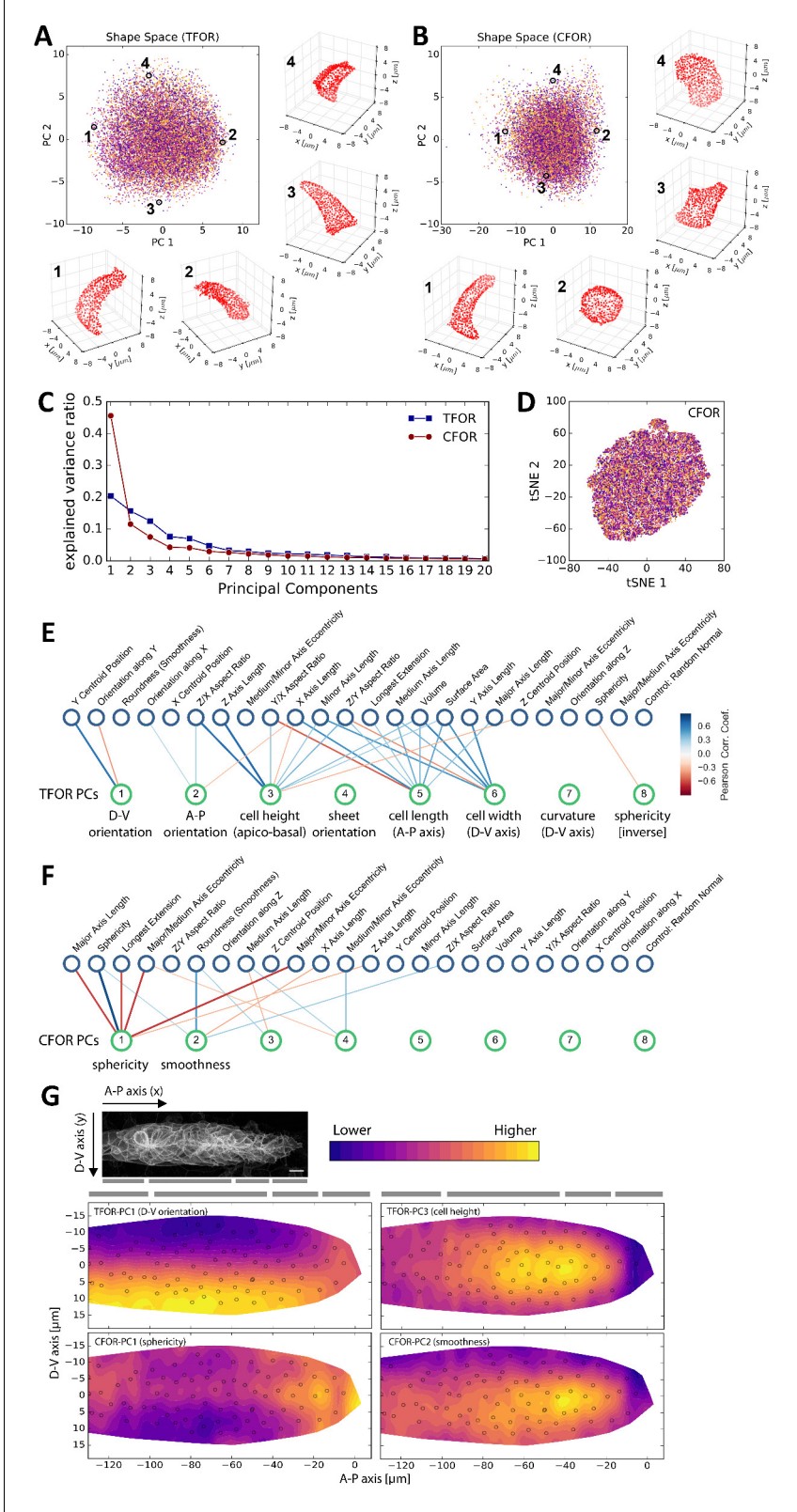

**Figure 4.** Analysis of the pLLP's Cellular Shape Space. (**A–B**) PCA plots of the tissue frame of reference (TFOR) and an cell frame of reference (CFOR) shape spaces of the pLLP. Each point represents a cell and each color represents a different primordium. Selected example cells are shown as point clouds, illustrating that meaningful properties are encoded in PCs, namely cell orientations (**A**) and cell sphericity and surface smoothness (**B**). (**C**) Explained variance ratios of principal components. (**D**) t-SNE embedding of the shape space showing the absence of obvious clusters as already seen

*Figure 4 continued on next page*

*Figure 4 continued*

with PCA in (**A–B**). Colors indicate different primordia as in (**A–B**). (**E–F**) Bigraph visualizations of correlations between principal components of the embedded space (bottom nodes) and a set of engineered features (top nodes). Any edge between two nodes indicates a correlation with Pearson's r > abs(0.3) and stronger edges indicate stronger correlations. A blue hue implies a positive and a red hue a negative correlation. These correlations together with manual inspection as shown in (**A–B**) allow the biological meaning of embedded features to be determined. (**G**) Consensus tissue maps of shape space PCs. The contour map represents the local average of PC values across all registered primordia. The small circles show the centroid positions of cells from a single example tissue to aid orientation. The gray bars indicate, from left to right, the deposition zone, follower zone, transition zone, and leader zone. pLLP shape features show varied patterns, including orientation along the D-V axis (TFOR-PC1), characteristic differences between leader and follower cells (TFOR-PC3, CFOR-PC2), and complicated patterns likely arising from the superimposition of different processes (TFOR-PC1).

pLLP do not cluster into discrete morphological groups (*Figure 4A–B,D*), implying a continuous shape spectrum between biologically distinct cells such as those at the tissue's leading edge and those at the center of assembled rosettes.

Embedded shape features are an unbiased encoding of shape without a pre-imposed biological meaning. It is thus interesting to ask what biological properties the most important shape features represent. To annotate the dimensions of the shape space with such biological properties, we correlated them against a curated set of simple engineered features (see materials and methods, and *Supplementary file 2*). Bigraph visualizations of these correlations reveal that the highest contributions to pLLP shape heterogeneity in the tissue frame of reference (TFOR) result from rotational orientation along different axes (PCs 1 and 2) as well as from absolute cell length in all three spatial dimensions (PCs 3, 5 and 6) (*Figure 4E*). Some of the PCs do not strongly correlate with any engineered features, implying that this information would have been lost without the use of unbiased feature embedding. Further manual inspection showed PC four to relate to additional rotational information for flat cells, PC seven to the curvature of cells along the tissue's dorso-ventral axis, and PC eight to cell sphericity. Overall, the dominance of orientation and size in the TFOR shape space is consistent with our previous observations on synthetic data (*Figure 3—figure supplement 2A*).

In the cell frame of reference (CFOR) (*Figure 4F*), which is invariant to size and rotation, cell shape heterogeneity is instead dominated by cell sphericity (PC 1), which accounts for nearly 50% of cell shape variation, with lesser contributions from cell surface smoothness (PC 2) and various minor factors that could not be annotated with a clear corresponding engineered feature. Cell sphericity and smoothness are thus important components of the pLLP's cellular architecture, a result that would have gone unnoticed without size- and rotation-invariant shape analysis. Intriguingly, both sphericity and smoothness are closely linked to cell surface mechanics, which have been little studied in the pLLP so far. The minor factors that remain unidentified in the CFOR shape space are likely mostly noise but PCs 3–5, which still explain a few percentage points of variance each (*Figure 4C*), may encode some meaningful shape information that is not obvious to the human eye and could only be retrieved using a computational approach.

The three-dimensional shape of cells in tissues is determined collectively via local cell interactions. Thus, one key advantage of image-based data is that extracted features are localized in the original spatial context of the tissue, allowing access to tissue-scale information that is lost in current single-cell omics approaches that require tissues to be dissociated. To analyze such tissue-scale patterns, we generated consensus maps of feature variation across primordia based on registered cell centroid positions (*Figure 4G*). As expected from our correlation analysis, features such as TFOR-PC1 (dorso-ventral orientation) are patterned along the corresponding axis of the tissue (top left panel). For TFOR-PC3 (cell height), the consensus map reveals an increase in cell height immediately behind the tissue's 'leading edge' (top right panel), which is consistent with the established notion that leader cells are flatter and follower cells transition into a more packed and columnar state (*Lecaudey et al., 2008*). This leader-follower progression is also reflected in CFOR-PC2 (cell surface smoothness), which displays a very similar pattern to TFOR-PC3 (*Figure 4G*, bottom right panel), possibly due to greater protrusive activity in leaders causing distortions in the cell's shape and thus reducing smoothness. These results provide unbiased quantitative support for previous observations on cell shape changes along the leader-follower axis but also indicate that these changes are continuous rather than discrete. This illustrates how our data-driven approach can provide a complete

picture of the patterns of key features across a tissue, which would be challenging to achieve by other means.

As established above, we found that CFOR-PC1 (sphericity) is a key component of the pLLP's cellular architecture. The corresponding consensus pattern is more intricate than the others (*Figure 4G*, bottom left panel), implying that different processes affect cell sphericity in the primordium. Cells at the front of the tissue register as more spherical due to reduced cell height, again as a consequence of the aforementioned leader-follower transition. The more intriguing pattern is that amongst followers there is a tendency for cells at the center of the tissue to be more spherical than those at the periphery, which manifests as a horizontal stripe in the consensus map. This pattern has not been described previously and may point toward an unknown mechanical aspect of pLLP self-organization.

Taken together, these results illustrate how data-driven exploratory analysis of the cellular shape space can reveal interesting patterns of shape heterogeneity that are indicative of specific mechanisms of organization.

## Data integration via machine learning on embedded features

A key strength of the ISLA-CBE pipeline is that it is not limited to embedding objects with continuous surfaces such as then cell plasma membrane. Instead, it is capable of embedding arbitrary intensity distributions, including those of the cytoskeleton, of organelles, or of any other labeled protein. Thus, ISLA-CBE can be used to analyze, and integrate, many aspects of cellular organization beyond morphology. This is of particular benefit when studying behaviors that are driven by the interaction of many cellular mechanisms, as is the case for tissue and organ shaping.

Our dataset encompasses many dual-color stacks containing both labeled membranes and one of several other cellular structures, including nuclei (*NLS-tdTomato*, *Figure 5A*), F-actin (*tagRFPt-UtrophinCH*, *Figure 5C*), and the Golgi apparatus (*mKate2-GM130*, *Figure 5E*) (see table in *Supplementary file 1* for a complete overview). For each of these structures we computed specific ISLA-CBE feature spaces (*Figure 5B,D,F*).

We sought to combine this intracellular information into an integrated quantitative atlas of cellular architecture. Such atlas integration of image data is usually performed in image space by spatial registration of samples (*Peng et al., 2011*; *Vergara et al., 2017*; *McDole et al., 2018*; *Cai et al., 2018*). However, registration requires stereotypical shapes that can be meaningfully overlaid via spatial transformations, so it is not readily applicable to developing tissues with non-stereotypical cellular positions. Machine learning can be employed as an alternative strategy for atlas mapping that side-steps this problem. Using a training set of dual-color images that contain both reference features (i.e. cell shape derived from the membrane label) and target features (i.e. the intracellular distribution of a specific protein), a machine learning model is trained to predict the target from the reference. The trained model can then be run on samples where only the reference (i.e. membranes) was imaged to generate predictions for the target (i.e. an intracellular protein). This strategy can be applied directly to microscopy images by means of deep convolutional neural networks (*Johnson et al., 2017*; *Christiansen et al., 2018*). However, such deep learning typically requires very large datasets that are usually unobtainable in developmental biology studies. Here, we instead chose to work in the embedded feature space, predicting ISLA-CBE embeddings of protein distributions based on ISLA-CBE embeddings of cell shape. This simplifies the problem such that smaller-scale machine learning algorithms become applicable.

We trained different classical (non-deep) machine learning models to predict the embedded feature spaces of subcellular structures based on the shape space. We optimized and validated this approach using grid search and cross validation (see materials and methods). Focusing on feature spaces in the tissue frame of reference (TFOR), we found that multi-output Support Vector Regression (SVR) yields good predictions across different conditions (*Figure 5—figure supplement 1*). We therefore used SVR prediction to generate a complete atlas spanning the embedded TFOR feature spaces of all available markers.

This atlas can be explored using the same visualizations shown previously for the shape space. As an example, we correlated the embedded features representing the Golgi apparatus (*mKate2-GM130*) with engineered features describing cell shape (*Figure 5G*). This shows that Golgi PCs 1 and 2 in the tissue frame of reference again relate to cellular orientation, whereas Golgi PC three correlates with cell height and a more columnar cell shape. Interestingly, the point cloud visualization

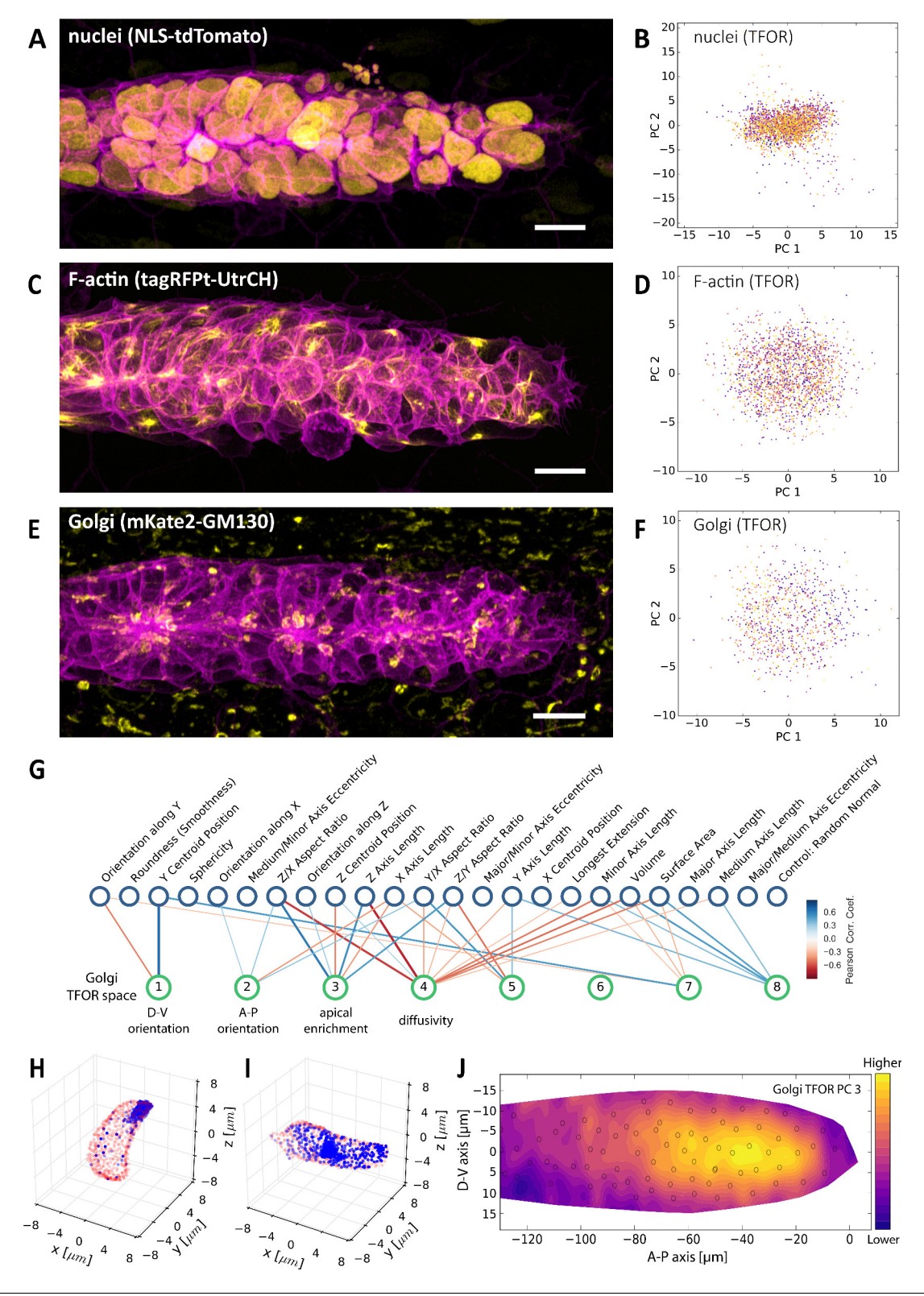

**Figure 5.** Multi-Channel Imaging, Embedding and Data Integration. (**A, C, E**) Maximum z-projections of two-color stacks showing the membrane in magenta and one of three subcellular structures in yellow. (**B, D, F**) Tissue frame of reference (TFOR) CBE embeddings corresponding to the three structures shown in A, C and E. The different colors of points indicate different primordia. The three structures are nuclei (N = 20, n = 2528) (**A–B**), F-actin (N = 19, n = 1876) (**C–D**) and the Golgi apparatus (N = 11, n = 866) (**E–F**). (**G**) Bigraph showing correlations between the Golgi's embedded

*Figure 5 continued on next page*

*Figure 5 continued*

features and our engineered cells shape features (see *Supplementary file 2*). The first two Golgi TFOR PCs match those found in the cell shape TFOR space (see *Figure 4E*) whereas PCs 3 and 4 are specific to the Golgi. For technical details see the legend of *Figure 4E*. (H–I) Point cloud renderings showing the distribution of Golgi signal (blue, membranes in red) in two example cells, one with a high value in the Golgi's TFOR PC 3 (H) and one with a low value (I), illustrating that PC three captures apical enrichment of the Golgi. (J) Consensus tissue map for Golgi PC 3 (apical enrichment), showing increased values behind the leader zone. For technical details see the legend of *Figure 4G*.

The online version of this article includes the following figure supplement(s) for figure 5:

**Figure supplement 1.** Evaluation of Machine Learning Algorithms for Feature Space Atlas Mapping.

reveals that Golgi PC three represents apical enrichment (*Figure 5H,I*) and the corresponding consensus tissue map shows that PC three is high in follower but not in leader cells (*Figure 5J*), indicating increased apical localization of the Golgi apparatus in followers. As apical Golgi localization is a hallmark of many epithelia (*Bacallao et al., 1989*), this finding provides unbiased quantitative support for a model where follower cells display increased epithelial character.

Besides subcellular protein distributions, changes in gene expression are key drivers of tissue organization. Indeed, one of the major goals of the field is to integrate gene expression data with cell and tissue morphology and behavior (*Battich et al., 2013*; *Lee et al., 2014*; *Satija et al., 2015*; *Karaiskos et al., 2017*; *Stuart et al., 2019*). We therefore investigated whether the machine learning approach to atlas construction we introduced here can be applied to any quantitative measurement that allows cell shape to be acquired simultaneously, including in-situ measurements of gene expression.

To this end, we performed single-molecule Fluorescence In-Situ Hybridization (smFISH) of *pea3* RNA (*Figure 6A*), a marker of FGFR signaling activity associated with the progression of follower cell development (*Aman and Piotrowski, 2008*; *Durdu et al., 2014*). We acquired 2-color stacks of smFISH probes and cell membranes from fixed samples (N = 31, n = 3149), employed automated spot detection to identify and count RNA molecules (*Figure 6—figure supplement 1A–E*) and embedded cell shapes with ISLA-CBE. We then used SVR to predict smFISH spot counts based on cell shape and location, finding that by combining all available information (the tissue frame of reference shape space, the cell frame of reference shape space and cell centroid coordinates) the trained model is able to account for 38.2 ± 1.9% of *pea3* expression variance (*Figure 6B*). The residual variance that could not be accounted for is due to high cell-to-cell heterogeneity in *pea3* expression among follower cells that does not seem to follow a clear spatial or cell shape-related pattern (*Figure 6C*). Nevertheless, we were able to recover the graded overall leader-follower expression pattern of *pea3* when running predictions across the entire atlas (*Figure 6D*). Our approach could therefore be used to superimpose at least the key patterns of gene expression in a tissue based on a set of in-situ labeling experiments, which in turn could potentially serve as a reference set to incorporate scRNA-seq data (*Stuart et al., 2019*).

Data integration based on common reference measurements such as cell shape counters one of the major shortcomings of current-day fluorescence microscopy, which is the limited number of channels and thus of molecular components that can be imaged simultaneously. The approach demonstrated here shows how embedded feature spaces and machine learning can be utilized to perform such data integration even in non-stereotypically shaped samples.

## Mapping of morphological archetypes facilitates biological interpretation

Feature embedding and atlas mapping enable the conversion of images into rich multi-dimensional numerical datasets. To take full advantage of this, biologically relevant patterns such as the relationships between different cell types, cell shapes and tissue context need to be distilled from these data and presented in a human-interpretable form. Accomplishing this in an automated fashion rather than by laborious manual data exploration remains a challenging problem in data science.

Classically, data interpretation involves relating data to established knowledge. This prompted us to look for ways to encode contextual knowledge quantitatively and use it to probe our atlas in a context-guided fashion. When asking biological questions about pLLP development, it is useful to distinguish different cell populations: the leader cells that are focused on migration, the cells at the

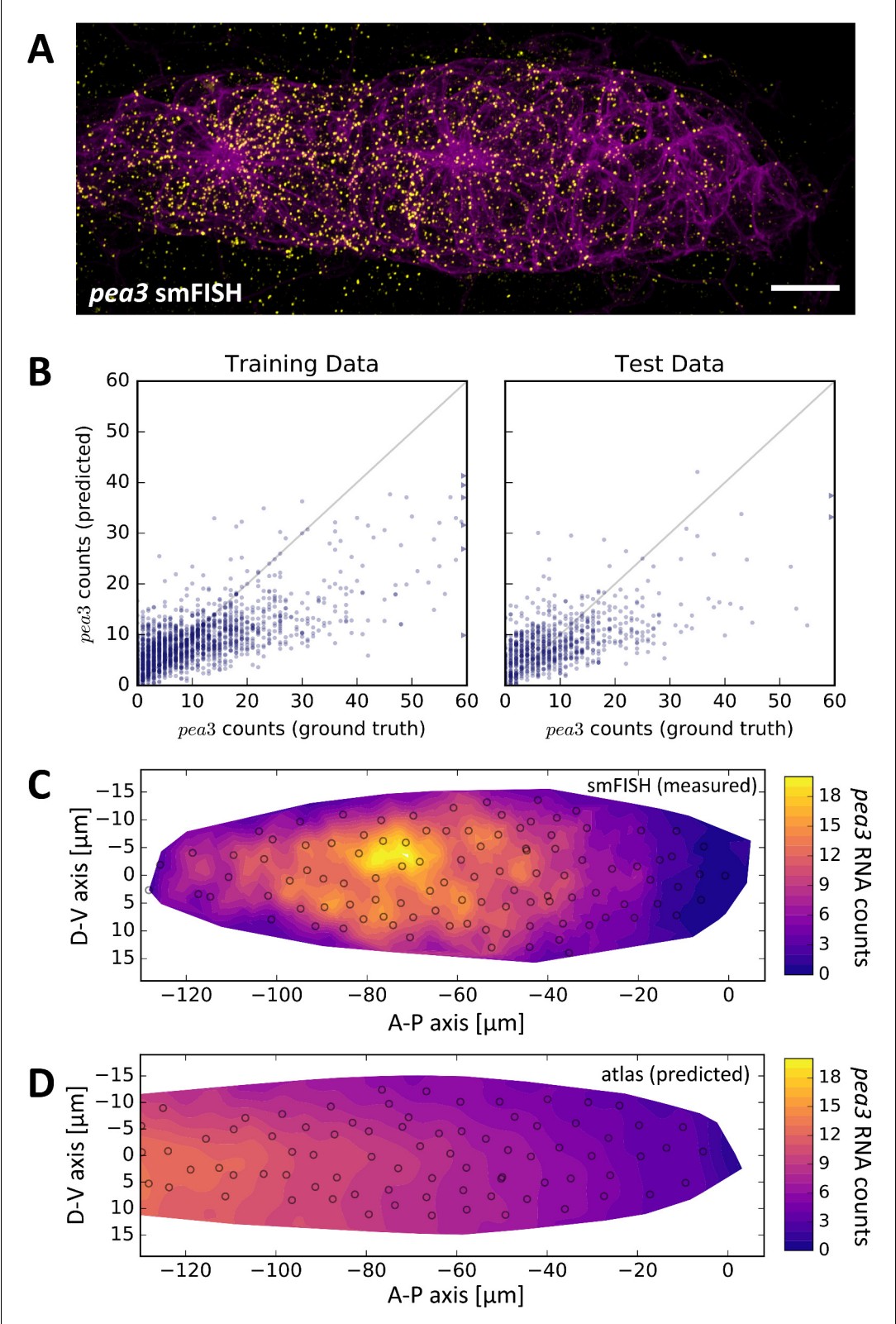

**Figure 6.** *pea3* smFISH as an Example of Data Integration Across Imaging Modalities. (**A**) Maximum z-projection of a two-color stack of *pea3* smFISH (yellow) and the *lyn-EGFP* membrane marker (magenta). Scale bar: 10 μm. (**B**) Results of SVR regression on *pea3* spot counts using TFOR and CFOR shape features as well as cell centroid coordinates of registered primordia as input. Each blue dot is a cell, the diagonal gray line reflects perfect prediction and blue arrows at the border point to outliers with very high spot counts. On training data, the regressor's explained variance ratio is

*Figure 6 continued on next page*

*Figure 6 continued*

0.462 ± 0.011, on previously unseen test data it achieves 0.382 ± 0.019. (C–D) Consensus tissue maps of *pea3* expression generated directly from the *pea3* smFISH dataset (C) or from the full atlas dataset based on SVR predictions of spot counts (D). Note that the prediction for the entire atlas preserves the most prominent pattern – the front-rear gradient across the tissue – but does not capture the noisy heterogeneity among follower cells observed in direct measurements.

The online version of this article includes the following figure supplement(s) for figure 6:

**Figure supplement 1.** Spot Detection and Cell Shape Embedding for *pea3* smFISH Data.

center of nascent rosettes that go on to form sensory hair cells later in development, the cells in the periphery of rosettes that will differentiate into so-called support and mantle cells and the cells in between rosettes (inter-organ cells) that will be deposited as a chain of cells between the maturing lateral line organs (*Grant et al., 2005*; *Hernández et al., 2007*; *Nogare et al., 2017*; *Figure 7A*). These four populations constitute simple conceptual archetypes that facilitate reasoning about the primordium's organization.

To map archetypes into the tissue's cellular shape space, we manually annotated cells that constitute unambiguous examples for each archetype in a subset of samples (N = 26, n = 624) (*Figure 7A*). We then again used machine learning – specifically a Support Vector Classifier (SVC) – with either tissue or cell frame of reference shape features as input to predict the class of all unlabeled cells. For both sets of input features, we found that the classifier was readily able to distinguish leader cells, central cells and peripheral cells, confirming that these manually chosen archetypes are indeed morphologically distinct (*Figure 7—figure supplement 1*). The inter-organ cells, however, were frequently misclassified as peripheral or central cells, indicating that at this developmental stage they are not substantially different in terms of shape from other follower cells (*Figure 7—figure supplement 1*). This illustrates that mapping of manually annotated information into the atlas intrinsically provides an unbiased quality control of biological pre-conceptions, as classification fails for archetypes that are not distinguishable from others.

Importantly, the SVC not only predicts categorical labels but also the probabilities with which each cell belongs to each class, a measure of how much a given cell resembles each archetype. This provides an entry point for human-interpretable visualization of atlas data. Here, we performed PCA on the prediction probabilities to arrive at an intuitive visualization wherein cells are distributed according to their similarity to each archetype (*Figure 7B,C*). In contrast to unsupervised dimensionality reduction of the shape space (see *Figure 4A,B,D*), this new visualization is readily interpretable for anyone with basic prior knowledge of pLLP organization.

One directly noticeable pattern is in the number of cells found in intermediate states between the leader, central and peripheral archetypes (*Figure 7C*). Intermediates between leaders and peripheral cells are common, as would be expected given the leader-follower axis along the length of the primordium. Similarly, intermediates between peripheral and central rosette cells are common, reflecting the continuous nature of rosette structure along the inside-outside axis. However, cells in an intermediate state between the leader and central rosette archetype are rare, possibly indicating that a direct transition between the two states does not frequently occur.

Any other data available at the single-cell level can be overlaid as a colormap onto the archetype visualization to take full advantage of its intuitive interpretability (*Figure 7D*). When doing so, previously observed patterns such as the increased cell height in followers – particularly in central cells – are clearly visible. Notably, the same pattern can be seen for PC 3 of the embedded protein distribution of F-actin, which is part of the atlas. This reflects apical enrichment of F-actin in follower cells, an important feature of rosette morphogenesis (*Lecaudey et al., 2008*). By contrast, plotting CFOR-PC1 (cell sphericity) shows that it is elevated in central rosette cells, especially in a population furthest from the peripheral archetype, a finding that was not predicted by previous studies. This is consistent with the spatial distribution seen in *Figure 4G* and reinforces the novel finding that cell sphericity exhibits a non-trivial pattern related to rosette organization. Interestingly, the pattern of CFOR-PC2 (cell surface smoothness) may represent the additive combination of the leader-follower and central-peripheral patterns, supporting a model where distinct mechanisms act to determine cell shape along these two axes of the tissue.

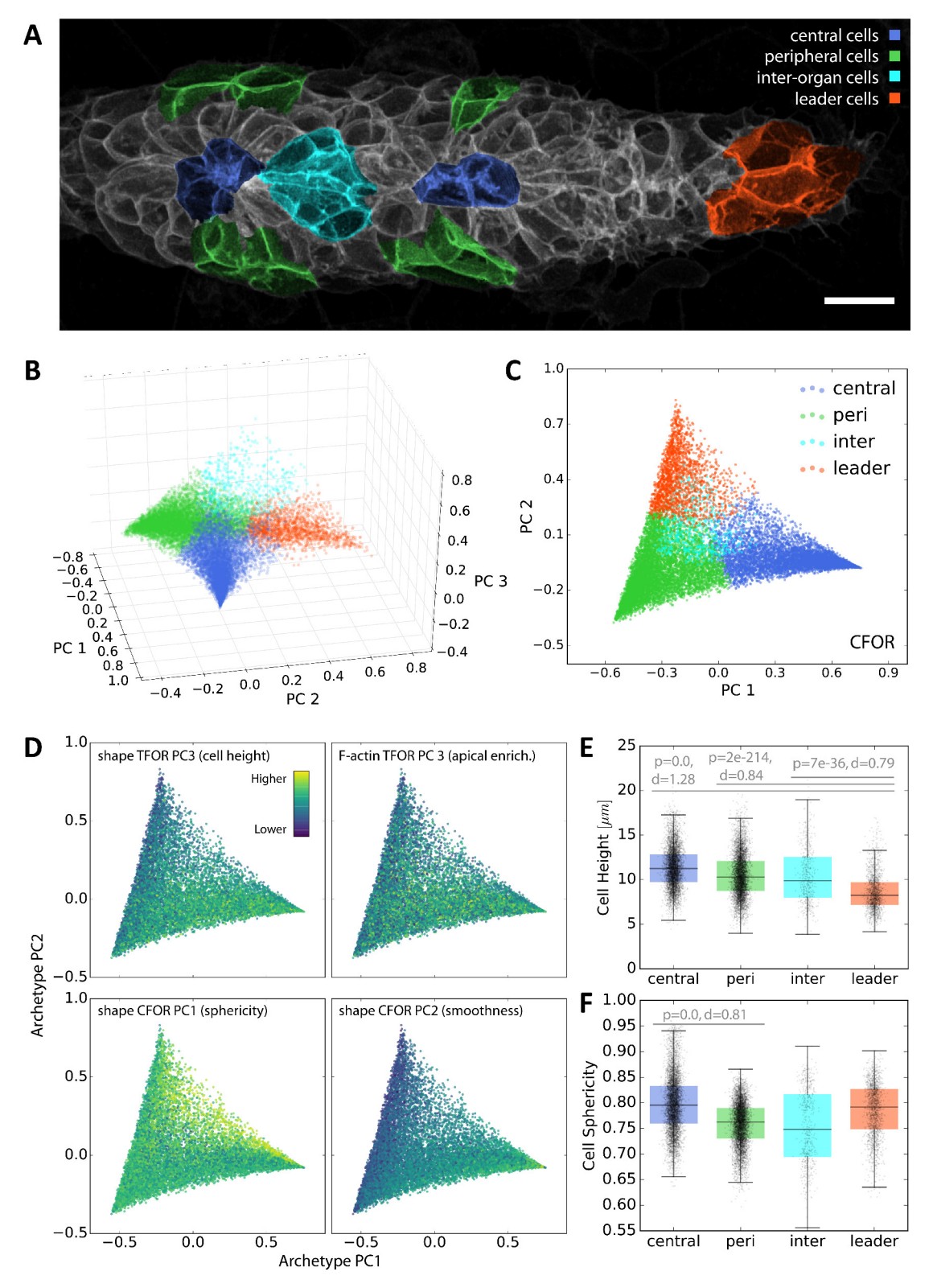

**Figure 7.** Context-Guided Visualization using Morphological Archetypes. (**A**) A maximum z-projected example stack with colors highlighting different conceptual archetypes in the pLLP that have been manually annotated. (**B**) A low-dimensional *archetype space* resulting from a PCA of the SVC prediction probabilities (with the SVC having been trained on CFOR shape features). Cells are placed according to how similar they are to each archetype, with those at the corners of the tetrahedron belonging strictly to the corresponding archetype and those in between exhibiting an

*Figure 7 continued on next page*

*Figure 7 continued*

intermediate morphology. (C) Since inter-organ cells are not morphologically distinct enough at this stage (see *Figure 7—figure supplement 1*), the archetype space can be reduced to 2D without much loss of information. (D) Scatter plots of the 2D archetype space with additional information from the cellular shape space and from the protein distribution atlas superimposed in color. (E–F) Boxplots showing data grouped by predicted archetype labels. This form of grouping allows statistical analysis, showing that leader cells are flatter than any other class of follower cells (E) and that central rosette cells are more spherical than peripheral rosette cells (F). Whiskers are 5th/95th percentiles, p-values are computed with a two-sided Mann-Whitney U-test, and Cohen's d is given as an estimate of effect size.

The online version of this article includes the following figure supplement(s) for figure 7:

**Figure supplement 1.** Evaluation of Morphological Archetype Prediction.

While these observations were made through an exploratory data-driven approach, the archetype labels also allow them to be checked with classical statistical hypothesis testing. Doing so substantiates both the previously known leader-follower difference in cell height (*Figure 7E*) and the novel finding that inner rosette cells are significantly more spherical than outer rosette cells (*Figure 7F*).

Despite its relative simplicity, our archetype-based approach showcases how the mapping of contextual knowledge onto a large and complicated dataset can facilitate its intuitive visualization and biological interpretation, leading to novel hypotheses that can be further tested experimentally.

## Discussion

The overarching goal of developmental biology is to explain how interactions at the molecular and cellular level lead to the formation of complex organisms. Data-driven approaches represent a major opportunity for advancement towards this aim, particularly in the search for holistic system-level understanding. However, seizing this opportunity requires the analysis of image-based multi-layered datasets spanning space, time, several scales and different modalities – a non-trivial task for which data science does not currently provide standardized solutions. Our data-driven analysis of cell and tissue architecture (*Figure 1*) addresses the three key challenges entailed in this task in the context of a typical developmental problem: data extraction, data integration, and data interpretation.

During data extraction, the rich information encoded in images must be pinpointed and transformed into a format more amenable to data science tools. This can be achieved by performing single-cell segmentation (*Figure 2*) and subsequently extracting numerical features describing each cell. Features can be generated through feature engineering, which has been employed successfully in a number of cases (*Wang et al., 2017*; *Viader-Llargués et al., 2018*). However, to gain a more complete and unbiased picture, feature embedding is preferable. We developed a novel feature embedding strategy inspired by classical geometric morphometrics, ISLA-CBE (*Figure 3*). ISLA allows arbitrary fluorescence intensity distributions to be converted to point clouds which in turn are embedded into a feature space with CBE. Previously described embedding strategies focus separately on shape (*Tweedy et al., 2013*; *Kalinin et al., 2018*) or on protein distributions (*Rajaram et al., 2012*; *Tweedy et al., 2013*; *Gut et al., 2018*), or require subcellular structures to be segmented into objects (*Peng and Murphy, 2011*; *Johnson et al., 2015*). Furthermore, shape-oriented methods usually require somewhat stereotypical objects that can be registered in order to achieve rotational invariance (*Pincus and Theriot, 2007*), a problem ISLA-CBE can solve through a simple pairwise distance transform (i.e. the cell frame of reference, CFOR). Recent work established neural network-based autoencoders as another highly promising approach for feature embedding (*Johnson et al., 2017*) but such deep learning methods tend to require large datasets for training and their internal workings are intractable (*Marcus, 2018*). ISLA-CBE has its own shortcomings, most notably that it is not easily reversible, meaning it is not readily possible to reconstruct a point cloud or a microscopy image from any point in the embedded feature space. Nevertheless, ISLA-CBE serves as a simple and broadly applicable tool for feature embedding of segmented cells.

Data integration across experiments is crucial to data-driven developmental biology because live microscopy is limited in the number of components that can be measured simultaneously. The value of generating such integrated data atlases has been demonstrated in several examples (*Peng et al., 2011*; *Vergara et al., 2017*), including a dynamic protein atlas of cell division (*Cai et al., 2018*) and an atlas of cell fate and tissue motion in the mouse post-implantation embryo (*McDole et al., 2018*). However, these examples employed registration in image space and thus relied on the stereotypical

shape of their samples, a prerequisite that cells in developing tissues do not necessarily conform to. Deep learning once again represents a promising alternative (*Johnson et al., 2017*; *Christiansen et al., 2018*; *Ounkomol et al., 2018*), though it remains to be demonstrated that it is applicable to the relatively small-scale datasets that can feasibly be produced in a developmental biology context. Here, we opted to use classical machine learning tools to perform multivariate-multivariable regression in order to map embedded feature spaces from different channels onto each other with cell shape as a common reference (*Figure 5*), an approach that could also easily be extended to gene expression data via smFISH (*Figure 6*). This simple and general solution comes with the downside that the resulting atlas cannot easily be viewed as an image overlay. Crucially, however, quantitative data analysis and visualization remain possible.

Finally, data interpretation is perhaps the most important and most challenging aspect. Even in fields leading in the adoption of data-driven methods, mining big datasets for human-readable mechanistic explanations remains a fundamental challenge (*Holzinger et al., 2014*; *Baker et al., 2018*). When aiming to make progress on this front, it is helpful to remind oneself that data on its own is not inherently meaningful; it can only be meaningfully interpreted in the context of an external conceptual framework (*Callebaut, 2012*; *Leonelli, 2019*). Guided by this notion, we sought a way of mapping contextual information onto the cellular shape space, allowing the entire dataset to be analyzed and visualized through a conceptual lens. Similar to recent work on the classification of cellular protrusions (*Driscoll et al., 2019*), we employed a machine learning classifier to map biologically meaningful cell archetype classes from a manually curated training set onto the entire dataset. We then utilized the prediction probabilities to generate a context-guided visualization that enables the biological interpretation of patterns across the entire atlas (*Figure 7*). Despite its relative simplicity, this approach represents a general strategy for bringing together data and context to facilitate interpretation.

Throughout our analysis, we found evidence that cells in the developing pLLP, from a morphological standpoint, do not fall into distinct groups but rather onto shape spectra between different states (*Figure 4A,B,D*). One such shape spectrum exists along the length of the primordium: leader cells are more flat and less defined by apicobasal polarity in their architectural features, whereas follower cells take on a more columnar architecture with the apical side as a focal point for cellular organization (*Figures 4G*, *5G–J* and *7D,E*). This provides data-driven support for previous observations that have led to a model where leaders exhibit mesenchyme-like features focused on driving migration and followers assume epithelial-like features geared toward rosette morphogenesis (*Nechiporuk and Raible, 2008*; *Lecaudey et al., 2008*; *Nogare et al., 2017*). However, it also emphasizes that these archetypical features are gradual in their nature rather than forming discrete clusters with sudden transitions. A second shape spectrum runs from the inside to the outside of assembling rosettes. As development proceeds, the cells along this spectrum will eventually differentiate into distinct cell types: the hair cells, support cells and mantle cells (*Hernández et al., 2007*; *Nogare et al., 2017*). Despite this well-known fate pattern, little attention has been paid thus far to the morphogenetic aspects of the inside-outside axis. Here, we found evidence that cell sphericity and cell surface smoothness, the two key shape factors of the pLLP's CFOR shape space, show inside-outside patterning as part of a more complicated multi-factorial pattern (*Figures 4F,G* and *7D,F*). Given the well-established link between these properties and effective cell surface tension (*Matzke, 1946*; *Lecuit and Lenne, 2007*), this leads us to the novel hypothesis that variations in adhesion and/or contractility could be the driving mechanism behind this pattern, as they would naturally lead to inside-outside sorting of cells in accordance with the Differential Interfacial Tension Hypothesis (DITH) (*Brodland, 2002*). Following this discovery in our exploratory analysis, further investigation will be required to test this idea and to dissect its biological implementation and function.

Collectively, the computational strategies presented in this study illustrate the potential of data-driven developmental biology to serve as a means to quantify, integrate and explore image-derived information. For this potential to be fully realized, future work will need to address three key points. First, a versatile open source software framework is needed that simplifies and standardizes handling of multi-layered biological datasets and thereby improves the accessibility of data science tools for developmental biologists. Second, further efforts are needed to facilitate biological interpretation of large multi-dimensional datasets, for instance by extending archetype mapping to other forms of contextual knowledge. Third, atlas datasets such as that of the pLLP need to be extended with

additional data and new modalities, including temporal dynamics (based on cell tracking), biophysical properties (such as cell surface tension) and the status of gene regulatory networks (derived e.g. from transcriptomics data). In the long term, we envision a comprehensive tissue atlas, a 'digital primordium', which can be mined for patterns and relationships across a wide range of experiments and modalities.

# Materials and methods

## Key resources table

| Reagent type (species) or resource | Designation | Source or reference | Identifiers | Additional information |
|---|---|---|---|---|
| Strain, strain background (*Danio rerio*) | *cldnb:lyn-EGFP* | DOI: 10.1016/j.devcel.2006.02.019 | ZFIN ID:ZDB-TG CONSTRCT-070117–15 | |
| Strain, strain background (*Danio rerio*) | *cxcr4b:NLS-tdTomato* | DOI: 10.1038/nature12635 | ZFIN ID:ZDB-TG CONSTRCT-141217–3 | |
| Strain, strain background (*Danio rerio*) | *Actb2:mKate2-Rab11a* | This paper | | made with tol2kit, see section 'Transgenic Fish Lines' |
| Strain, strain background (*Danio rerio*) | *LexOP:CDMPR-tagRFPt* | This paper | | made with tol2kit, see section 'Transgenic Fish Lines' |
| Strain, strain background (*Danio rerio*) | *LexOP:B4GalT1(1-55Q)-tagRFPt; cxcr4b:LexPR* | This paper | | made with tol2kit, see section 'Transgenic Fish Lines' |
| Strain, strain background (*Danio rerio*) | *atoh1a:dtomato* | DOI: 10.1016/j.ydbio.2010.02.017 | ZFIN ID:ZDB-TG CONSTRCT-100928–4 | |
| Strain, strain background (*Danio rerio*) | *6xUAS:tagRFPt-UtrCH* | This paper | | made with tol2kit, see section 'Transgenic Fish Lines' |
| Strain, strain background (*Danio rerio*) | *cxcr4b:LexPR* | doi: 10.1038/nature13852 | ZFIN ID:ZDB-TG CONSTRCT-170105–4 | LexOP chemically inducible driver line |
| Strain, strain background (*Danio rerio*) | ETL GA346 | doi: 10.1073/pnas.0903060106 | | Gal4-UAS driver line |
| Recombinant DNA reagent | *sp6:mKate2-Rab5a* (plasmid) | This paper | | made with tol2kit, see section 'Transgenic Fish Lines' |
| Recombinant DNA reagent | *sp6:mKate2-GM130(rat)* (plasmid) | doi: 10.1242/jcs.026849 | | made with tol2kit, see section 'Transgenic Fish Lines' |
| Sequence-based reagent | CDMPR FOR (*D. rerio*, full-length) | This paper | PCR primers | GGGGACAAGTTTGTACAAAAAAG CAGGCTGGATGTTGCTGT CTGTGAGAATAATCACT |
| Sequence-based reagent | CDMPR REV (*D. rerio*, full-length) | This paper | PCR primers | GGGGACCACTTTGTACAAGAAAGCTGGGT CCATGGGAAGTAAATGGTCATCTCTTTCCTC |
| Sequence-based reagent | B4GalT1 FOR (*D. rerio*, 1-55Q) | This paper | PCR primers | GGGGACAAGTTTGTACAAAAAAGCAGGC TGGATGTCGGAGTCGGTGGGATTCTTC |
| Sequence-based reagent | B4GalT1 REV (*D. rerio*, 1-55Q) | This paper | PCR primers | GGGGACCACTTTGTAC AAGAAAGCTGGGTCT TGTGAATTAACCATATCA GAGATAAATGAAATGTGTCG |
| Sequence-based reagent | Rab5a FOR (*D. rerio*, full-length) | This paper | PCR primers | GGGGACAGCTTTCTTGTACAAAGTGGCTA TGGCCAATAGGGGAGGAGCAACAC |
| Sequence-based reagent | Rab5a REV (*D. rerio*, full-length) | This paper | PCR primers | GGGGACAACTTTGTATAATAAAGTTG CTTAGTTGCTGCAGCAGGGGGCT |

*Continued on next page*

*Continued*

| Reagent type (species) or resource | Designation | Source or reference | Identifiers | Additional information |
|---|---|---|---|---|
| Sequence-based reagent | Rab11 FOR (*D. rerio*, full-length) | This paper | PCR primers | GGGGACAGCTTTCTTGTACAAAGTGGCT ATGGGGACACGAGACGACGAATACG |
| Sequence-based reagent | Rab11 REV (*D. rerio*, full-length) | This paper | PCR primers | GGGGACAACTTTGTATAATAAAGTTG CCTAGATGCTCTGGCAGCACTGC |
| Sequence-based reagent | Quasar 670-conjugated Stellaris smFISH probes | LGC, Biosearch Technologies | smFISH probes | Targeted at *D. rerio* pea3 gene, see section 'smFISH: Fixation, Staining and Imaging' |
| Commercial assay or kit | RNeasy Mini Kit | Qiagen | Cat#:74106 | |
| Commercial assay or kit | QIAshredder | Qiagen | Cat#:79654 | |
| Commercial assay or kit | SuperScript III Reverse Transcriptase | Thermo Fisher Scientifc | Cat#:18080044 | |
| Commercial assay or kit | mMESSAGEmMACHINE SP6 Transcription Kit | Thermo Fisher Scientific | Cat#:AM1340 | |
| Commercial assay or kit | MultiSite Gateway Pro | Invitrogen | Cat#:12537 | |
| Chemical compound, drug | LysoTracker Deep Red | Thermo Fisher Scientific | Cat#:L12492 | |
| Chemical compound, drug | RU486 | Sigma-Aldrich | Cat#:M8046 | |
| Chemical compound, drug | N-phenylthiourea (PTU) | Sigma-Aldrich | Cat#:P7629 | |
| Chemical compound, drug | Tricaine (MESAB) | Sigma-Aldrich | Cat#:A5040 | |
| Software, algorithm | fiji | doi: 10.1038/nmeth.2019 | RRID:SCR_002285 | imagej.net/Fiji |
| Software, algorithm | Anaconda | Anaconda, Inc | RRID:SCR_018317 | www.anaconda.com |
| Software, algorithm | python 2.7 | Python Software Foundation | RRID:SCR_008394 | www.python.org/psf |
| Software, algorithm | ISLA | This paper | github.com/WhoIsJack/data-driven-analysis-lateralline (*Hartmann, 2020*; copy archived at https://github.com/elifesciences-publications/data-driven-analysis-lateralline) | See section 'Data and Code Availability' |
| Software, algorithm | CBE | This paper | github.com/WhoIsJack/data-driven-analysis-lateralline | See section 'Data and Code Availability' |

## Animal handling

Zebrafish (*Danio rerio*, RRID:ZFIN_ZDB-GENO-060919-1) were grown, maintained and bred according to standard procedures described previously (*Westerfield, 2000*). All experiments were performed on embryos younger than 3dpf, as is stipulated by the EMBL internal policy 65 (IP65) and European Union Directive 2010/63/EU. Live embryos were kept in E3 buffer at 27–30°C. For experiments, pigmentation of embryos was prevented by treating them with 0.002% N-phenylthiourea (PTU) (Sigma-Aldrich, St. Louis, US-MO) starting at 25hpf. For mounting and during live imaging, embryos were anaesthetized using 0.01% Tricaine (Sigma-Aldrich, St. Louis, US-MO).

## Transgenic fish lines

All embryos imaged carried the membrane marker *cldnb:lyn-EGFP* (*Haas and Gilmour, 2006*), which was used for single cell segmentation. In addition, different subsets of the main dataset carried one of the following red secondary markers: *cxcr4b:NLS-tdTomato* (nuclei) (*Donà et al., 2013*), *Actb2: mKate2-Rab11a* (recycling endosomes), *LexOP:CDMPR-tagRFPt* (trans-Golgi network and late endosomes), *LexOP:B4GalT1(1-55Q)-tagRFPt* (*trans*-Golgi), *6xUAS:tagRFPt-UtrCH* (F-actin) or *atoh1a:dtomato* (transcriptional marker for hair cell specification) (*Wada et al., 2010*). Furthermore, in two subsets of the data a red marker was injected as mRNA, namely *mKate2-GM130(rat)* (*cis*-Golgi) (*Pouthas et al., 2008*) and *mKate2-Rab5a* (early endosomes). Finally, one subset of samples was treated with 1 µM LysoTracker Deep Red (Thermo Fisher Scientific, Waltham, US-MA) in E3 medium with 1% DMSO for 90 min prior to imaging. The exact composition of the main dataset is tabulated in *Supplementary file 1*.

To drive expression of the UAS construct, those fish additionally carried the Gal4 enhancer trap ETL GA346 (*Distel et al., 2009*). To drive expression of LexOP constructs (*Emelyanov and Parinov, 2008*), those fish carried *cxcr4b:LexPR* (*Durdu et al., 2014*) and were treated with 10 µM of the progesterone analogue RU486 (Sigma-Aldrich, St. Louis, US-MO) from 25hpf.

Capped mRNA was produced by IVT using the mMESSAGEmMACHINE SP6 Transcription Kit (Thermo Fisher Scientific, Waltham, US-MA) according to the manufacturers' instructions and was injected at 250 ng/µl into embryos at the 1 cell stage.

The following plasmids were generated by MultiSite Gateway Pro Cloning (Invitrogen, Waltham, US-MA) based on the Tol2kit (*Kwan et al., 2007*): *LexOP:CDMPR-tagRFPt* (with *cry:mKate2* transgenic marker), *LexOP:B4GalT1(1-55Q)-tagRFPt* (with *cry:mKate2* transgenic marker), *Actb2:mKate2-Rab11a* (with *clmc2:GFP* transgenic marker), *6xUAS:tagFRPt-UtrCH* (with *cry:mKate2* transgenic marker), *sp6:mKate2-GM130(rat)* and *sp6:mKate2-Rab5a*. tagRFPt (*Shaner et al., 2008*) and UtrCH (*Burkel et al., 2007*) were kindly provided by Jan Ellenberg and Péter Lénárt, respectively. Zebrafish CDMPR, B4GalT1, Rab5a and Rab11a were cloned by extraction of total RNA from dechorionated 48hpf zebrafish embryos using the RNeasy mini kit (Qiagen, Hilden, Germany) and QIAshredder (Quiagen, Hilden, Germany) according to manufacturer's instructions, followed by reverse transcription with SuperScript III Reverse Transcriptase (Thermo Fisher Scientific, Waltham, US-MA) using both random hexamers and oligo-dT simultaneously, according to the manufacturer's instructions, and finally amplification of genes of interest from cDNA with the following oligonucleotides (Sigma-Aldrich, St. Louis, US-MO) (template-specific region underlined):

CDMPR FOR: GGGGACAAGTTTGTACAAAAAAGCAGGCTGG<u>ATGTTGCTGTCTGTGAGAATAATCACT</u>
CDMPR REV: GGGGACCACTTTGTACAAGAAAGCTGGGTC<u>CATGGGAAGTAAATGGTCATCTCTTTCCTC</u>
B4GalT1 FOR: GGGGACAAGTTTGTACAAAAAAGCAGGCTGG<u>ATGTCGGAGTCGGTGGGATTCTTC</u>
B4GalT1 REV: GGGGACCACTTTGTACAAGAAAGCTGGGTC<u>TTGTGAATTAACCATATCAGAGATAAATGAAATGTGTCG</u>
Rab5a FOR: GGGGACAGCTTTCTTGTACAAAGTGGCT<u>ATGGCCAATAGGGGAGGAGCAACAC</u>
Rab5a REV: GGGGACAACTTTGTATAATAAAGTTGC<u>TTAGTTGCTGCAGCAGGGGGCT</u>
Rab11a FOR: GGGGACAGCTTTCTTGTACAAAGTGGCT<u>ATGGGGACACGAGACGACGAATACG</u>
Rab11a REV: GGGGACAACTTTGTATAATAAAGTTGC<u>CTAGATGCTCTGGCAGCACTGC</u>

## High-Resolution live imaging

Embryos were manually dechorionated with forceps at 30-34hpf and anaesthetized with 0.01% Tricaine (Sigma-Aldrich, St. Louis, US-MO), then transferred into 1% peqGOLD Low Melt Agarose (Peqlab, Erlangen, Germany) in E3 containing 0.01% Tricaine and immediately deposited onto a MatTek Glass Bottom Microwell Dish (35 mm Petri dish, 10 mm microwell, 0.16–0.19 mm coverglass) (MatTek Corporation, Ashland, US-MA). No more than 10 embryos were mounted in a single dish. A weighted needle tool was used to gently arrange the embryos such that they rest flatly with their lateral side directly on the glass slide. After solidification of the agarose, E3 containing 0.01% Tricaine was added to the dish. Embryos were imaged at 32-36hpf, when the pLLP was located above the posterior half of the embryo's yolk extension.

The microscope used for imaging was the Zeiss LSM880 with AiryScan technology (Carl Zeiss AG, Oberkochen, Germany), henceforth LSM880. High-resolution 3D stacks (voxel size: 0.099 μm in xy, 0.225 μm in z) were acquired with a 40 × 1.2 NA water objective with Immersol W immersion fluid (Carl Zeiss, Oberkochen, Germany). Imaging in AiryScan FAST mode (*Huff, 2016*) with a piezo stage for z-motion and bi-directional scanning allowed acquisition times for an entire volume to be lowered to approximately 20 s (40 s for dual-color stacks using line switching). Deconvolution was performed using the LSM880's built-in 3D AiryScan deconvolution with 'auto' settings.

Note that optimal image quality could only be achieved by adjustment of the stage to ensure that the cover glass is exactly normal to the excitation beam. For each dish we imaged, we used 633 nm reflected light and line scanning to get a live view of the cover glass interface, which allowed us to manually adjust the pitch of the stage to be completely horizontal. This process was repeated for both zx and zy line scans.

## smFISH: Fixation, Staining and Imaging

Single molecule Fluorescence In-Situ Hybridization (smFISH) was performed according to standard protocols (*Durdu et al., 2014*; *Raj et al., 2008*) using previously published Quasar 670-conjugated Stellaris smFISH probes (LGC, Biosearch Technologies, Hoddesdon, UK) designed to target *pea3* mRNA, listed below.

Briefly, embryos were fixed overnight in 4% PFA in PBS-T (PBS with 0.1% Neonate-20) at 4°C, then rinsed three times in PBS-T and subsequently permeabilized with 100% methanol overnight at −20°C. Embryos were rehydrated with a methanol series (75%, 50%, 25% Methanol in PBS-T, 5 min per step) and rinsed three times with PBS-T. The yolk was manually removed using forceps. Next, samples were pre-incubated with hybridization buffer (0.1 g/ml deyxtrane sulfate, 0.02 g/ml RNase-free BSA, 1 mg/ml *E. coli* tRNA, 10% formamide, 5x SSC, 0.1% Neonate-20 in ddH2O) at 30°C for 30 min and subsequently hybridized with *pea3* probe solution (0.1 μM in hybridization buffer) at 30°C overnight in the dark. After probe removal, embryos were stained with DAPI (1:1000) in washing buffer (10% formamide, 5x SSC, 0.1% Neonate-20 in ddH2O) for 15 min at 30°C and finally kept in washing buffer for 45 min at 30°C.

Stained embryos were mounted on glass slides using VECTASHIELD HardSet Antifade Mounting Medium (Vector Laboratories, Burlingame, US-CA) and imaged immediately to prevent loss of signal due to photobleaching. Imaging was performed with a 63 × 1.4 NA oil immersion objective on the LSM880 in FAST mode with 488 nm and 639 nm excitation lasers. Stacks were acquired using 8x averaging with 0.187 μm z-spacing and a pixel size of 0.085 μm, then deconvolved with the built-in 3D AiryScan deconvolution on 'auto' settings.

The following smFISH probes were used:

1: aaggaagacggacagaggca, 2: ctgtgttttaatgagctcca, 3: cttaaccgtttgtggtcatt,
4: ccatccatcttataatccat, 5: agtataaggcacttgctggt, 6: atttccttgcgacctattag,
7: tcaacagtctatttaggggc, 8: atgtatttccttttgtcgc, 9: aagaggtcttcagattcctg,
10: cctgaagttggcttaaatcc, 11: ggaacttgagcttcggtgag, 12: aacaaactgctcatcgctgt,
13: cactgagttctctgagtgaa, 14: ttcttaatcttcacaggcgg, 15: tagctgaagctttgcttgtg,
16: tcataggcactggcgtaaag, 17: ctggacatgagctcttagat, 18: ttgggggaataatgctgcat,
19: tgagggtggattcatatacc, 20: cggaagggaacctggaactg, 21: agagtgttgccgatggaaac,
22: tgctgaggaggataaggcaa, 23: ccatgtactcctgcttaaag, 24: tcctgtttgaccatcatatg,
25: caggttcgtaagtgtagtcg, 26: tgtgatggtacatggatggg, 27: aaacatgtagccttcactgt,
28: tggcacaacacgggaatcat, 29: tcacctcaccttcaaatttc, 30:accttcacgaaacacactgc,
31: tagttgaagtgagccacgac, 32: gaagggcaaccaagaactgc, 33: atgcgatgaagtgggcattg,
34: atgagtttgaattccatgcc, 35: ttgtcatagttcatggctgg, 36: gtaacgcaaagagcgactca,
37: ttttgcataattcccttctc, 38: aggttatcaaagcttctggc, 39: cgctgattgtcgggaaaagc,
40: gttgacgtagcgctcaaatt, 41: aagaaactccctcatcgagg, 42: tacatgtagcctttggagta,
43: aaaggagaatgtcggtggca, 44: gtggtaaactgggatgggaa, 45: atacaagaggatggggtggg,
46: gaatgcagagtccctaatga, 47: agataggcctcagaagtgag, 48: gcaatctcttgaaccacagt.

## Software development stack

The software for this study was developed using the Anaconda distribution (Anaconda, Inc, Austin, US-TX) of python 2.7.13 (64-bit) (Python Software Foundation, Beaverton, US-OR) (*Van Rossum, 1995*).

The following scientific libraries and modules were used: numpy 1.11.3 (*Travis and Oliphant, 2006*) and pandas 0.19.2 (*McKinney, 2010*) for numerical computation, scikit-image 0.13.0 (*van der Walt et al., 2014*) and scipy.ndimage 2.0 (*Jones and Oliphant, 2001*) for image processing, scikit-learn 0.19.1 (*Pedregosa and Varoquaux, 2011*) for machine learning, matplotlib 1.5.1 (*Hunter, 2007*) and seaborn 0.7.1 (*Waskom et al., 2016*) for plotting, networkx 1.11 (*Hagberg et al., 2008*) for graph-based work, tifffile 0.11.1 (*Gohlke, 2016*) for loading of TIFF images, and various scipy 1.0.0 (*Jones and Oliphant, 2001*) modules for different purposes. Parallelization was implemented using dask 0.15.4 (*Team, 2016*).

Jupyter Notebooks (jupyter 1.0.0, notebook 5.3.1) (*Kluyver et al., 2016*) were utilized extensively for prototyping, workflow management and exploratory data analysis, whereas refactoring and other software engineering was performed in the Spyder IDE (spyder 3.2.4) (*Raybaut et al., 2018*). Version control was managed with Git 2.12.2.windows.2 (*Torvalds and the Git contributors, 2018*) and an internally hosted GitLab instance (GitLab, San Francisco, US-CA).

## Image preprocessing

Following AiryScan 3D deconvolution with 'auto' settings on the LSM880, images were converted to 8bit TIFF files using a custom macro for the Fiji distribution (*Schindelin et al., 2012*) of ImageJ 1.52 g (*Schneider et al., 2012*). The minimum and maximum values determining the intensity range prior to 8bit conversion were selected manually such that intensity clipping is avoided. Care was taken to apply the same values to all samples of a given marker to ensure consistency.

Samples with the *cxcr4b:NLS-tdTomato* nuclear label exhibited a degree of bleed-through into the *lyn-EGFP* membrane label channel. To prevent this from interfering with single-cell segmentation, we employed a linear unmixing scheme in which the contribution of *NLS-tdTomato* ($C$, the contaminant image) is removed from the green channel ($M$, mixed image), resulting in the cleaned membrane channel ($U$, unmixed image). Our approach assumes that the signal in $M$ is composed according to *Equation 1*, implying that $U$ can be retrieved by subtraction of an appropriate contamination term (*Equation 2*).

$$M = U + a \cdot C \tag{1}$$

$$U = M - a \cdot C \tag{2}$$

To compute the optimal bleed-through factor $a$ we minimized a custom loss function (*Equation 3*), which is essentially simply the Pearson Correlation Coefficient ($PCC$) of the contaminant image $C$ and the cleaned image $U$ given a particular candidate factor $a_i$. To ensure that unreasonably high values of $a$ are punished, we centered the values of the cleaned image onto their mean and converted the result to absolute values, causing overly unmixed regions to start correlating with $C$ again.

$$loss = PCC(C, \ |M - a_i \cdot C - mean(M - a_i \cdot C)|) \tag{3}$$

We found that this approach robustly removes *NLS-tdTomato* bleed-through, producing unmixed images that could be segmented successfully.

## Single-Cell segmentation

3D single-cell segmentation was performed on membrane-labeled stacks acquired, deconvolved and preprocessed as detailed in the sections above.

The pipeline for segmentation consists of the following steps, applied sequentially:

1. 3D median smoothing with a cuboid $3 \times 3 \times 3$ vxl structural element to reduce shot noise.
2. 3D Gaussian smoothing with σ = 3 pxl to further reduce noise and smoothen structures.
3. Thresholding to retrieve a binary mask of foreground objects (i.e. the membranes).
   To automatically determine the appropriate threshold, we use a custom function inspired by a semi-manual approach for spot detection (*Raj et al., 2008*). Starting from the most frequent value in the image histogram as a base threshold, we iteratively scan a limited range of positive offsets (usually 0 to 10 in steps of 1, for slightly lower-quality images 0 to 40 in steps of 2) and count the number of connected components in the inverse of the binary mask resultant

from applying each threshold. This roughly represents the number of cell bodies in the stack that are fully enclosed in cell membranes at a given threshold. We consider the threshold producing the largest number the best option and use it to generate the final membrane mask.

4. Removal of disconnected components by morphological hole filling.
5. Labeling of connected components on the inverted membrane mask. This ideally yields one connected component per cell, i.e. the cytoplasm.
6. Removal of connected components smaller than 1'000 voxels (artifacts) and re-labeling of connected components larger than 1'000'000 voxels as background objects.
7. Watershed expansion using the labeled connected components as seeds and the smoothed input image as topography (with additional 3D Gaussian smoothing on top of steps 1 and 2, with $\sigma = 3$ pxl). The background objects surrounding the primordium are also expanded.
8. Assignment of the zero label to background objects and removal of any objects disconnected from the primordium by retaining only the single largest foreground object.

We manually optimized the parameters of this pipeline for our data by inspecting the output during an extensive set of test runs.

Finally, we also manually double-checked all segmentations and discarded rare cases where more than approx. 10% of cells in a stack had been missed or exhibited under- or oversegmentation.

## Point cloud sampling with ISLA

*Intensity-biased Stochastic Landmark Assignment* (ISLA) (*Figure 3A–C*, *Figure 3—figure supplement 1A*) was applied to cropped-out bounding boxes of single segmented cells. To capture cell shape, the 6-connected inner hull of the binary segmentation mask was used as input image for ISLA. To capture intensity distributions, voxels outside the segmentation mask were set to zero and a simple background subtraction was performed to prevent landmarks from being assigned spuriously due to background signal. The background level was determined as the mean intensity within the masked cell and was subtracted from each voxel's intensity value, with resulting negative values set to zero.

With the inputs so prepared, voxel intensities were normalized such that their sum equals one by dividing each by the sum of all. Then, landmarks were assigned by considering the normalized voxel intensities as the probabilities of a multinomial distribution from which 2000 points were sampled (with replacement). This number was determined through test runs in which we found that, for volumes of our size, diminishing returns set in around 500 points and using more than 2000 points no longer improved the performance of various downstream analyses.

Following ISLA sampling, landmark coordinates were scaled from pixels to microns to account for anisotropic image resolution.

Note we recently published another study that utilizes a simplified version of ISLA for some of the data analysis, albeit in a very different way from how it is used here (*Wong et al., 2020*).

## Point cloud transformation into TFOR and CFOR

To place cells in a matched *Tissue Frame of Reference* (TFOR) prior to feature embedding (*Figure 3—figure supplement 1B*, left route), primordia were aligned using a simple PCA-based approach that does not require full image registration. To this end, 3000 landmarks were sampled from a given primordium's binary overall segmentation mask using ISLA and a modified PCA was applied to the resulting point cloud. Given that the pLLP's longest axis is always its front-rear axis and the shortest axis is always the apicobasal axis, PCA transformation snaps primordia that had been acquired at a slight angle into a consistent frame of reference. Our modification ensured that 180° flipping could not occur in this procedure. To complete the alignment, the primordial point clouds were translated such that the frontal-most point becomes the coordinate system's origin. Finally, the ISLA point clouds extracted from individual cells as described in the previous paragraph were transformed using the same PCA, thus matching their orientation to the common tissue frame of reference.

To create a *Cell Frame of Reference* (CFOR) that is invariant to size, rotation and handedness (*Figure 3—figure supplement 1B*, right route), point cloud volumes were first normalized such that the sum of the magnitudes of all centroid-to-landmark vectors is 1. Second, cellular point clouds were cast into a pairwise distance (PD) representation. In the PD space, each point of the cloud is no longer characterized by three spatial coordinates but instead by the distances to every other point

of the cloud. This representation is rotationally invariant but also extremely high-dimensional (an LxL array, where L is the number of landmarks). To reduce dimensionality, only the 10th, 50th and 90th percentiles of all pairwise distances for each point were chosen to represent that point (resulting in an Lx3 array), which we reasoned would encode both local and global relative spatial location.

## Feature embedding with CBE

To determine reference cluster centers for *Cluster-Based Embedding* (CBE) (*Figure 3D*, *Figure 3— figure supplement 1B*), point clouds from multiple samples were centered on their respective centroids and overlaid. K-means clustering was performed on this overlaid cloud (using scikit-learn's *MiniBatchKMeans* implementation) with *k = 20*. The resulting cluster centers were used as common reference points for the next step.

Several measures were taken to improve the robustness and performance of this cluster detection. First, individual cellular point clouds were downsampled from 2000 points to 500 points prior to being overlaid, using k-means clustering with *k = 500* clusters, the centers of which were used as the new landmarks. Second, not all available cells were used in the overlay. Instead, a representative random subset of primordia (at least 10, at most 25) was selected and only their cells were used in the overlay. The resulting cluster centers were used as reference points across all available samples. Third, the entire overlaid point cloud was downsampled using a density-dependent downsampling approach inspired by *Qiu et al., 2011* (see below), yielding a final overlaid cloud of at most 200'000 points, which allowed reference cluster centers to be computed reasonably efficiently.

Density-dependent downsampling was performed using a simplified version of the algorithm described by *Qiu et al., 2011*. First, the local density (*LD*) at each point is found, which is defined as the number of points in the local neighborhood, i.e. a sphere whose radius is the median pairwise distance between all points multiplied by an empirically determined factor (here 5). Next, a target density (*TD*) is determined, which in accordance with Qiu et al. was set to be the third percentile of all local densities. Now, points are downsampled such that the probability of keeping each point is given by *Equation 1*. If necessary, the resulting downsampled distribution is further reduced by random sampling in order to reach the maximum of 200'000 points.

$$p(keep\_cell\_i) = \begin{cases} 1, & if \ LD_i < TD \\ \frac{TD}{LD_i}, & otherwise \end{cases} \tag{4}$$

Density-dependent downsampling was chosen to avoid cases where high-density agglomerations of landmarks in a particular region accumulate multiple clusters and thus deplete lower-density regions of local reference points; density-dependent downsampling preserves the overall shape of the overlaid point cloud whilst reducing local density peaks.

Following the determination of common reference points, CBE proceeds by extracting features describing the local landmark distribution around said reference points for each separate cellular point cloud. Several such features were implemented, including the number of landmarks in the local neighborhood of each reference point, the number of landmarks assigned to each reference point by the k-means clustering itself, the local density of landmarks at each reference point determined by a Gaussian Kernel Density Estimate (KDE), and either the magnitude or the components of the vector connecting each reference point to the centroid of its 25 nearest neighbors. The results were similar across all of these approaches but we ultimately chose to proceed with the last entry in the above list (the vector components; see *Figure 3D* for a simple 2D example) based on its inclusion of some additional directional information.

The feature extraction described above yields an *n-by-3k* latent feature space, where *n* is the number of cells and *k* the number of shared reference clusters (here *k = 20*). As a final step, this space was transformed by Principal Component Analysis (PCA) to bring relevant variation into focus and reduce dimensionality.

In addition to CBE, an alternative embedding based on the moments of the TFOR or CFOR point clouds was also generated as a comparably simple baseline. We computed the 1 st raw moments (*Equation 5*), the 2nd centralized moments (*Equation 6*) and the 3rd to 5th normalized moments (*Equation 7*) (55 features in total) and once again used PCA to arrive at a compact and expressive feature space.

$$rM_{1[ijk]} = mean\left(C_z \cdot i + C_y \cdot j + C_x \cdot k\right) \tag{5}$$

$$cM_{2[ijk]} = mean\left(\left(C_z - rM_{1[100]}\right)^i \cdot \left(C_y - rM_{1[010]}\right)^j \cdot \left(C_x - rM_{1[001]}\right)^k\right) \tag{6}$$

$$nM_{m[ijk]} = \frac{cM_{m[ijk]}}{std\left(C_z - rM_{1[100]}\right)^i \cdot std\left(C_y - rM_{1[010]}\right)^j \cdot std\left(C_x - rM_{1[001]}\right)^k} \tag{7}$$

In **Equations 5, 6, 7**, $C_d$ is the array of all point cloud coordinates along the spatial dimension $d$, $M_m$ is the set of raw ($rM_m$), centralized ($cM_m$) or normalized ($nM_m$) moments of the $m$-th order, and $[i, j, k]$ includes all combinations of length 3 drawn from the integer range $[0, \ldots, m]$ that satisfy $i + j + k = m$. All operations are element-wise except $mean(\ldots)$ and $std(\ldots)$, which compute the mean and standard deviation across a given array.

## Synthetic point cloud generator

In order to benchmark the capability of CBE for latent feature extraction, we required a gold standard dataset to test whether CBE recovers known latent parameters underlying a population of shape objects. We thus wrote a generator to automatically create a large synthetic dataset of cell-like point clouds. This generator functions in four stages:

1. Generate a spine (five parameters)
   The spine is defined by its height along the z dimension (i.e. the apicobasal axis), by an offset, which is the distance and angle by which the end point of the spine is shifted in xy compared to the starting point, and by a connecting line between endpoints, which is a 2nd degree polynomial.
2. Generate the cell surface (nine parameters)
   The surface of the cell is determined based on its distance $d$ to the spine as a function of $z$. We used the logit-normal as this distance function on the grounds that it has only two parameters, can be asymmetric, monomodal or bimodal, crosses (0, 0) and (1, 0), and has a range $d > 0$ within the domain $0 > z > 1$. For each cell, three different logit-normals were used to describe the surface at three uniformly spaced angles around the spine. Additionally, each of those functions was independently scaled by multiplication with another parameter.
3. Sampling of point clouds (0 parameters)
   The actual surface point cloud to be used in feature embedding was generated by sampling 2000 points with the following scheme: first randomly sample a position along $z$ and an angle, then determine the corresponding radius by linear interpolation between the logit-normal values of the two adjacent angles at which they are defined. Finally, angle and radius are converted to the Cartesian coordinate system.
4. Centering, scaling and rotation (three parameters)
   Finally, point clouds are centered on their centroid and cell size is adjusted by multiplication with a scaling parameter, followed by rotation using a homogeneous transformation matrix with two angle parameters.

All in all, this generator requires 17 parameters of which one modifies only size (not shape) and two modify only rotation. For each synthesized cell, the specific parameter values for the generator were sampled from normal or uniform distributions with hyperparameters set based on empirical experimentation such that a varied population of cell shapes is generated and unreasonably deformed shapes are avoided.

## Evaluation of CBE on synthetic point clouds

Using the generator described above, we synthesized a set of $n = 20'000$ cellular point clouds and embedded them using both CBE and the moment-based alternative embedding strategy. We then tested how well the values of the 17 known generative parameters could be retrieved from the embedded feature spaces using different scikit-learn implementations of multivariate-multivariable regression, specifically k-Nearest Neighbor (kNN) regression, multi-output linear Support Vector Regression (l-SVR) and multi-output radial basis function kernel Support Vector Regression (r-SVR). Optimal hyperparameters for the two SVRs were determined using cross-validated grid search with

*GridSearchCV*, scanning 5 orders of magnitude surrounding the defaults of C (penalty) and epsilon as well as gamma (RBF kernel coefficient) for r-SVR.

This synthetic experiment showed that cell size and orientation largely obscure other shape features if CFOR-normalization is not performed (*Figure 3—figure supplement 2A*). Furthermore, features generated by CBE enable similar or better predictive performance than moments-based features regardless of the method used for prediction (*Figure 3—figure supplement 2B*). Interestingly, kNN performs markedly better on CBE-embedded spaces, indicating that CBE more meaningfully encodes point cloud similarity as local neighborhood in the embedded space.

## Feature engineering: Extraction of simple shape measures

We extracted various explicitly engineered features from entire tissues, segmented cell volumes, and segmented cell point clouds. The features used in this study are listed and briefly described in *Supplementary file 2*.

## Multi-Channel atlas prediction

To construct the multi-channel atlas, we used the secondary markers present in many of our samples (see *Supplementary file 1*), embedded them with ISLA and CBE (see the corresponding sections above) and then trained multivariate-multivariable regressors to predict those embeddings based on the corresponding cell shape embeddings as input features. All embeddings were standardized and PCA-transformed prior to machine learning and only the first 20 PCs were considered. Predictions were performed from shape TFOR to secondary marker TFOR and from shape CFOR to secondary marker CFOR.

Because expression of the secondary markers was sometimes heterogeneous across the primordium, only cells with a secondary marker intensity above the 33rd percentile were used as training data. Furthermore, to prevent rare extreme outliers (resulting from segmentation artifacts) in either the shape or secondary marker feature space from affecting the prediction, such anomalies were detected and removed from the training data using the isolation forest algorithm (*sklearn.ensemble. isolation_forest*) with *contamination = 0.05*.

To select the best machine learning model, the following regressors were tested using 3-fold cross-validation based on scikit-learn's *ShuffleSplit* function: k-nearest neighbors regression (*sklearn. neighborsKNeighborsRegressor*), random forest regression (*sklearn.ensemble.RandomForest Regressor*), elastic net regression (*sklearn.linear_model.ElasticNet*), Lasso regression (*sklearn. linear_-model.Lasso*), a multi-layer perceptron (*sklearn.neural_network.MLPRegressor*), and a support vector regressor with an RBF-kernel (*sklearn.svm.SVR*). Relevant hyperparameters were optimized on the *NLS-tdTomato* nuclear marker using a 5-fold cross-validated grid search (*sklearn.model_selection. GridSearchCV*) of 5 orders of magnitude surrounding the scikit-learn defaults. Performance was evaluated across different secondary channels and latent feature embeddings (*Figure 5—figure supplement 1A*), with the primary aim being high explained variance but also giving some consideration to computational efficiency (training and prediction time). After finding that TFOR predictions perform substantially better than CFOR predictions, possibly because the TFOR shape space incorporates more relevant information for intracellular marker prediction and/or because CFOR spaces contain more noise and more specific information that is challenging to predict, we focused further analysis on TFOR for the time being. We selected the RBF SVR as the regressor of choice because of its consistently high performance.

For atlas prediction across the entire dataset, the SVR model was trained for each secondary channel (using the corresponding best hyperparameter set) on all available data for that channel and then applied to predict that channel's embedded space for all other cells.

## smFISH: Spot detection and analysis

Single-cell segmentation and cell shape feature embedding was performed on the *pea3* smFISH dataset in the same fashion as for the live imaging dataset, resulting in 3'149 cells from 31 samples.

The *blob_log* spot detector from scikit-image was used (*skimage.blob.blob_log*) to identify smFISH spots (*Figure 6—figure supplement 1A–D*). Parameters were optimized by manual testing and by scanning different values for the *threshold* and *min_sigma* parameters, arriving ultimately at *min_sigma = 1, max_sigma = 4, num_sigma = 10, threshold = 0.22, overlap = 0.5*, and

*log_scale = False*, which produced average counts per cell that are reasonably consistent across different primordia (*Figure 6—figure supplement 1E*) and closely approximate previously reported *pea3* smFISH spot counts in the pLLP (*Durdu et al., 2014*). However, we also found that optimal settings can vary between different smFISH experiments and thus recommend parameter optimization and careful evaluation of the results for every experiment. Furthermore, in this dataset the spot detector failed to detect a reasonable number of spots in some exceptional samples, so we excluded primordia with a mean count per cell of less than two spots from further analysis, leaving 2906 cells from 29 primordia in the dataset.

Because smFISH must be performed on fixed samples, we checked for fixation effects on cell shape, which would make cross-predictions with the live sample atlas more challenging. In bulk comparisons of key shape features (*Figure 6—figure supplement 1F–H*) we found no significant differences between live and fixed samples. However, such bulk comparisons may miss subtle fixation effects, making the development of methods to quantify, pinpoint and correct such effects an important future goal.

We trained an RBF-kernel SVR to perform multivariate regression of *pea3* smFISH counts based on as much other information available about each cell as possible (*Figure 6B*, *Figure 6—figure supplement 1E*), namely a combined feature space incorporating the first 10 shape TFOR PCs, the first 10 shape CFOR PCs, and the x, y and z coordinates of cell centroids in TFOR. Hyperparameters C (penalty), epsilon, and gamma (RBF kernel coefficient) were again optimized using *GridSearchCV*, whereas *ShuffleSplit* was used to randomly split data into training and test data for evaluation as shown in *Figure 6B*.

## Prediction and visualization of morphological archetypes

The four archetypes were manually annotated in 26 primordia (*Figure 7A*) using Fiji's multi-point selection tool, yielding 93 leader cells, 241 outer rosette cells, 182 inner rosette cells and 108 between-rosette cells (624 cells in total). Only the clearest examples of the respective archetypes were labeled.

Cell archetype prediction was performed using scikit-learn's Support Vector Classifier (SVC) with a Radial Basis Function (RBF) kernel, using either TFOR- or CFOR-embedded cell shapes as input features. We optimized the SVC hyperparameters for each embedding using scikit-learn's *GridSearchCV* with 5-fold cross-validation, testing whether or not to use feature standardization, whether or not to use PCA and keep the first 15, 30 or 50 PCs, as well as screening 5 orders of magnitude surrounding the scikit-learn default values for C (penalty) and gamma (RBF kernel coefficient). The resulting best estimator was used for all further training and prediction.

The confusion matrices in *Figure 7—figure supplement 1* were produced by splitting the annotated cells into a training set (436 cells) and a test set (188 cells) using scikit-learn's *StratifiedShuffleSplit* to balance labels. Predictions for the entire atlas dataset were generated following training with all 624 manually annotated cells.

The archetype space was constructed by inferring the classification probabilities for each class (using *sklearn.svm.SVC.predict_proba*) and performing a PCA on them. The 3D and 2D visualizations in *Figure 7* were then generated by plotting the first three or two principal components, respectively.

## Image rendering and expanded view of segmentation

Fiji's *Straighten* tool was used to align angled samples with the main image axes for the purpose of illustration in *Figures 1*, *2*, *5*, *6* and *7* and in *Figure 2—video 1*, *2*, *3* but never as part of an image analysis pipeline. Maximum projections were created with Fiji whereas 3D videos were rendered with Imaris 7.7.2 (Bitplane, Belfast, UK).

The expanded view of the segmented pLLP (*Figures 1B* and *2C*) was generated by first determining the centroids of each segmented cell and then shifting them apart by scaling of their x and y coordinates by a single user-specified factor. The cells were then pasted into an appropriately scaled empty image stack at the new centroid locations, leaving them shifted apart uniformly but not individually rescaled or otherwise transformed. A python implementation of this approach called *tissueRipper* is available under the MIT open source license on GitHub at github.com/WhoIsJack/tissueRipper.

## Correlation heatmaps and bigraphs

The corresponding bigraphs (*Figures 4E,F* and *5G*) were generated using a custom plotting function based on the networkx module. The edges were colored according to the signed value of the Pearson correlation coefficient and sized according to its magnitude. Edges with an absolute correlation coefficient smaller than 0.3 were omitted. The nodes of the engineered features were sorted to reduce edge crossings and group similar nodes, which was achieved by minimizing the following custom loss function:

$$loss = \sum_{i=0}^{f_E} \sum_{j=0}^{f_L} \left| \frac{C_i}{f_E} - \frac{j}{f_E} \right| \cdot \left| pcc(E_i, L_j) \right| \tag{8}$$

where $f_E$ and $f_L$ are the number of engineered and latent features, respectively. $C$ is the current sort order of the engineered features, that is a permutation of the integer interval $[0, f_E]$. Finally, $\left| pcc(E_i, L_j) \right|$ is the absolute Pearson correlation coefficient of the values of the i-th engineered feature and the *j*-th latent feature. In essence, this loss function is the sum of all Euclidean rank distances between engineered and latent features, weighted by their corresponding absolute Pearson correlation coefficients. Minimization was performed by random shuffling of the sort order and retaining only shuffles that reduced the loss until no change was observed for 2000 consecutive shuffles.

Note that since the sign of principal components is not inherently meaningful, we flipped it for shape TFOR-PCs 1, 3, 5, six and shape CFOR-PC one across all analyses presented in this study to ensure that PCs positively correlate with their most defining engineered feature(s), facilitating discussion of the results.

## Tissue consensus maps

Consensus maps of feature variation (*Figures 4G*, *5J* and *6C,D*) were based on an overlay of TFOR centroid positions of cells across all relevant samples.

The cut-off for the consensus tissue outline was determined by computing the local density of these overlaid centroids using scipy's Gaussian kernel density estimation with default settings. Regions with densities below 10% of the range between the minimum and maximum density were considered outside of the primordium and are shown as white in the plot.

The local consensus feature values were computed by applying a point cloud-based Gaussian smooth across the individual cells' feature values, using the 0.5th percentile of all pairwise distances between centroids as $\sigma$. This smoothed distribution was then plotted as a *tricontourf* plot with matplotlib using up to 21 automatically determined contour levels.

## Statistical analysis

Generally, we use N to refer to the number of embryos/primordia and n to the number of cells.

Statistical significance for comparisons between two conditions was estimated without parametric assumptions using a two-tailed Mann-Whitney U test (*scipy.stats.mannwhitneyu* with keyword argument *alternative='two-sided'*) with Bonferroni multiple testing correction (*Haynes, 2013*) where appropriate. We considered *p>0.01* as not statistically significant.

Significance tests with large sample sizes such as those encountered during single-cell analysis tend to indicate high significance regardless of whether the difference between populations is substantive or technical (*Sullivan and Feinn, 2012*), which is why we also report estimates of effect size for any comparison that is statistically significant. Effect sizes were estimated using *Cohen, 1977*. The resulting values can be described as *no effect* (d ≈ 0.0), a *small effect* (d ≈ 0.2), a *medium effect* (d ≈ 0.5) or a *large effect* (d ≈ 0.8) (*Cohen, 1977*).

## Materials availability

Requests for experimental resources and reagents should be directed to and will be fulfilled by Darren Gilmour (darren.gilmour@imls.uzh.ch).

## Data and code availability

All raw and processed data are available freely and openly via the Image Data Resource repository (idr.openmicroscopy.org) under accession number idr0079. Code is available under the MIT open source license on GitHub at github.com/WhoIsJack/data-driven-analysis-lateralline. Note that we aim to update the core algorithms to python 3 and make them available as a readily reusable module in the near future. Inquiries regarding data and code should be directed to Jonas Hartmann (jonas.m.hartmann@protonmail.com).

## Acknowledgements

We thank Sabine Görgens and Andreas Kunze for their support with fish and lab maintenance, respectively. We thank the EMBL Advanced Light Microscopy Facility (ALMF) and the UZH Center for Microscopy and Image Analysis (ZMB) for maintenance of and assistance with microscopes. We thank Alejandra Guzman Herrera for generating the *Act2b:mKate2-Rab11a* line. We thank Francesca Peri and Stefano De Renzis for kindly providing temporary lab space. We thank Christian Tischer and Marvin Albert for helpful discussion on image analysis and numerical computation. We thank Stefano De Renzis, Daniel Krueger, Marvin Albert and Andrew Kennard for critical reading of the manuscript. JH and EG were supported by the EMBL International PhD Programme (EIPP), M.W. was supported by an EMBO Long-Term Fellowship and the EMBL Interdisciplinary Postdoc (EIPOD) Program under Marie Curie COFUNDII Actions. The Gilmour lab was supported by the European Molecular Biology Laboratory (EMBL), the University of Zurich (UZH) and Swiss National Science Funds Grant 31003A_176235.

## Additional information

### Funding

| Funder | Grant reference number | Author |
| --- | --- | --- |
| European Molecular Biology Laboratory | International PhD Programme | Jonas Hartmann Elisa Gallo |
| European Molecular Biology Organization | ALTF 205-2015 | Mie Wong |
| H2020 Marie Skłodowska-Curie Actions | COFUNDII - EMBL Interdisciplinary Post-doc (EIPOD) Program | Mie Wong |
| European Molecular Biology Laboratory | | Darren Gilmour |
| Schweizerischer Nationalfonds zur Förderung der Wissenschaftlichen Forschung | 31003A_176235 | Darren Gilmour |
| University of Zurich | | Darren Gilmour |

The funders had no role in study design, data collection and interpretation, or the decision to submit the work for publication.

### Author contributions

Jonas Hartmann, Conceptualization, Data curation, Software, Formal analysis, Validation, Investigation, Visualization, Methodology, Writing - original draft, Writing - review and editing; Mie Wong, Elisa Gallo, Investigation, Writing - review and editing; Darren Gilmour, Conceptualization, Resources, Supervision, Funding acquisition, Project administration, Writing - review and editing

### Author ORCIDs

Jonas Hartmann https://orcid.org/0000-0002-5600-8285
Elisa Gallo https://orcid.org/0000-0003-2203-6787
Darren Gilmour https://orcid.org/0000-0001-7613-090X

## Ethics

Animal experimentation: All experiments on zebrafish embryos were performed between 0-3 days post-fertilization according to the rules of the European Molecular Biology Laboratory, following the guidelines of the European Commission, Directive 2010/63/EU.

## Decision letter and Author response

Decision letter https://doi.org/10.7554/eLife.55913.sa1
Author response https://doi.org/10.7554/eLife.55913.sa2

# Additional files

### Supplementary files

• Supplementary file 1. Table of Dataset Composition. The number of primordia (N) and of segmented single cells (n) that was used in the analysis for each combination of fluorescent labels. The numbers shown do not include the eight samples that were discarded due to low segmentation quality.

• Supplementary file 2. Table of Engineered Features. Brief descriptions of engineered features that were extracted from each segmented cell and used for analysis, in particular to assign biological properties to embedded features extracted with ISLA-CBE.

• Transparent reporting form

### Data availability

All raw and processed data is openly available via the Image Data Resource repository (https://idr.openmicroscopy.org) under accession number idr0079.

The following dataset was generated:

| Author(s) | Year | Dataset title | Dataset URL | Database and Identifier |
|---|---|---|---|---|
| Hartmann J, Wong M, Gallo E, Gilmour D | 2020 | idr0079-hartmann-lateralline | https://idr.openmicroscopy.org/search/?query=Name:idr0079 | Image Data Resource, idr0079 |

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
