## [Decision Letter]

**Acceptance summary:**

This work develops an image analysis pipeline for extracting and analyzing cell-based data using unsupervised machine learning to find correlations between cell shape, orientation, position, and gene expression. Timelapse imaging of the developing zebrafish lateral line organ is used as an example of this approach. The authors develop an approach to statistically represent cell shape in both a cell-based and tissue-based reference frame to construct a feature space for machine learning. This representation is used to build an atlas integrating multiple cellular markers and linking them to cell type archetypes. This tour de force pioneering study is expected to open an avenue to a more effective analysis of "data-rich" microscopy data, which provides one of the most important yet challenging windows on unfolding of the embryogenesis and organogenesis.

**Decision letter after peer review:**

Thank you for submitting your article "An image-based data-driven analysis of cellular architecture in a developing tissue" for consideration by *eLife*. Your article has been reviewed by two peer reviewers, and the evaluation has been overseen by a Reviewing Editor and Marianne Bronner as the Senior Editor The following individual involved in review of your submission has agreed to reveal their identity: Ajay B. Chitnis (Reviewer #1).

The reviewers have discussed the reviews with one another and the Reviewing Editor, who also read the manuscript and has drafted this decision to help you prepare a revised submission.

Summary:

This work develops an image analysis pipeline for extracting and analyzing cell-based data using unsupervised machine learning to find correlations between cell shape, orientation, position, and gene expression. Timelapse imaging of the zebrafish lateral line is used as an example of this approach. The authors develop an approach to statistically represent cell shape in both a cell-based and tissue-based reference frame to construct a feature space for machine learning. This representation is used to build an atlas integrating multiple cellular markers, such as F-actin or Golgi organelle, predict smFISH data and, is linked to cell type archetypes. This tour de force pioneering study is expected to open an avenue to very effective analysis of "data-rich" microscopy data, which provides one of the most important yet challenging windows on unfolding of the embryogenesis and organogenesis.

Essential revisions:

Whereas the two reviewers and the managing editor were in agreement about the significance and broader applicability of the methods you developed, they differed in their view on the manuscript presentation.

One reviewer thought that the paper is well laid out, and despite its complexity, the logic and significance of each step in the analysis made fairly accessible to a diverse reader audience. The authors describe the challenges of data extraction, data integration and data interpretation. They address these challenges by describing how high-resolution 3D images collected with AiryScan fast mode confocal microscopy were first segmented with automation. Then a four-step process was implemented to extract and integrate information from these images.

However, the second reviewer thought that this paper is a bit of a "diamond in the rough". It was not easy to understand what the authors were doing after reading the title, Abstract, Introduction, or first section of the Results. And it was not until the reviewer read the remaining results that they got excited that this was a novel, general, and up-and-coming approach. Having read the manuscript, the reviewing editor agrees with this reviewer that revising the manuscript with the goal of increasing its clarity and accessibility would significantly improve its impact.

Therefore, we recommend that you should re-work the first half of the paper considering the following points with the goal of better communicating your approach to a biology audience. Be 1) more specific, 2) less jargony, and 3) describe goals with multiple words. Moreover, more detail about the patio-temporal details of what was imaged should be added in the Results section. The methodological approach is described as a tool for studying organogenesis, a highly dynamic process. Yet, only from the Materials and methods section one can learn that embryos at 30-34 hpf (and thus practically a single time point) have been imaged. Please, indicate whether a primordium at a specific position in the embryo was imaged. Also, the authors could more clearly describe the feature integration experimental setup, especially whether integration of signals like F-actin, or Golgi apparatus requires simultaneous imaging of the membrane signal. Consulting the manuscript with non-specialists would be also advised during the revision process.

In addition, as reviewers point out, one is a little disappointed that despite the elegant analysis, what the authors discovered, with some minor exceptions, for the most part confirmed what we knew to a large extent about morphogenesis in the pLLP. Therefore, perhaps more exploration of differences between cells along the inside out axis rather than along the longitudinal or DV axis might have provided some additional important distinctions between cells in terms of how they might correlate with Nuclei, Actin or golgi distribution. Such new insight would underscore the power of the approach and its impact in the field.

---

## [Author Response]

Essential revisions:Whereas the two reviewers and the managing editor were in agreement about the significance and broader applicability of the methods you developed, they differed in their view on the manuscript presentation.One reviewer thought that the paper is well laid out, and despite its complexity, the logic and significance of each step in the analysis made fairly accessible to a diverse reader audience. The authors describe the challenges of data extraction, data integration and data interpretation. They address these challenges by describing how high-resolution 3D images collected with AiryScan fast mode confocal microscopy were first segmented with automation. Then a four-step process was implemented to extract and integrate information from these images.However, the second reviewer thought that this paper is a bit of a "diamond in the rough". It was not easy to understand what the authors were doing after reading the title, Abstract, Introduction, or first section of the Results. And it was not until the reviewer read the remaining results that they got excited that this was a novel, general, and up-and-coming approach. Having read the manuscript, the reviewing editor agrees with this reviewer that revising the manuscript with the goal of increasing its clarity and accessibility would significantly improve its impact.Therefore, we recommend that you should re-work the first half of the paper considering the following points with the goal of better communicating your approach to a biology audience. Be 1) more specific, 2) less jargony, and 3) describe goals with multiple words. Moreover, more detail about the patio-temporal details of what was imaged should be added in the Results section. The methodological approach is described as a tool for studying organogenesis, a highly dynamic process. Yet, only from the Materials and methods section one can learn that embryos at 30-34 hpf (and thus practically a single time point) have been imaged. Please, indicate whether a primordium at a specific position in the embryo was imaged. Also, the authors could more clearly describe the feature integration experimental setup, especially whether integration of signals like F-actin, or Golgi apparatus requires simultaneous imaging of the membrane signal. Consulting the manuscript with non-specialists would be also advised during the revision process.

We have made a range of adjustments aimed at improving clarity and better describing the motivations behind various steps in our analysis throughout the first half of the paper. Regarding jargon, we in some cases opted to keep the technical terminology for the benefit of specialist readers and added further information to clarify its meaning for non-specialists. Keeping the standard terminology while providing accessible explanations has the added benefit of hopefully making general data-science literature less intimidating to those non-specialist readers who'd like know about this important emerging field. After following the recommendation to consult non-specialists, we believe that this revised manuscript is now sufficiently clear and well-contextualized for a general audience. However, if the reviewers or the reviewing editor see further room for improvement, we will be happy to incorporate specific suggestions.

Regarding timing, embryos were mounted at 30-34hpf and imaged at 32-36hpf with the primordium located above the posterior half of the yolk extension. We have clarified this in the Results and Materials and methods sections. The internal organization of the pLLP is known to be relatively consistent throughout its migration (excepting the start and end), so our analysis focuses on this particular time frame as a means of investigating tissue organization. As indicated in the Discussion section, our approach could readily be extended to dynamic data, which will be the subject of future studies.

Finally, we have clarified in the data integration section that simultaneous imaging is required in order to produce training data for machine learning.

In addition, as reviewers point out, one is a little disappointed that despite the elegant analysis, what the authors discovered, with some minor exceptions, for the most part confirmed what we knew to a large extent about morphogenesis in the pLLP. Therefore, perhaps more exploration of differences between cells along the inside out axis rather than along the longitudinal or DV axis might have provided some additional important distinctions between cells in terms of how they might correlate with Nuclei, Actin or golgi distribution. Such new insight would underscore the power of the approach and its impact in the field.

We thank the reviewers for this helpful and constructive suggestion. Based on this feedback, we revisited the key patterns found within our atlas with the aim of adding further information on the central-peripheral axis of pLLP organization. In the original manuscript, we tended to highlight the 'known' features we were able to extract, as this helped validate our new unbiased method. However, there were indeed a number of novel insights and these are now emphasized in the revised version.

Regarding novel insights resulting from additional markers specifically, such as nuclei and actin, we found that the structures currently featured in the atlas are either dominated by leader-follower variation or add little new information beyond what is already captured by cell shape (nuclei and actin in particular tend to simply recapitulate overall cell shape variation). However, we believe this is offset by the exciting general demonstration that additional markers of interest can be integrated so seamlessly into our shape atlas and we're confident that this 'multiplexing' capability will lead to novel insights in other tissue contexts.

We have followed the helpful suggestion to further explore differences along the inside-outside axis. In Figure 7D, we now include CFOR-PC2 (cell surface smoothness), which shows an interesting pattern that further reinforces our finding that a previously unknown mechanism acting through cell surface mechanics may play a role in rosette morphogenesis. We made modifications to the corresponding paragraphs in the Results and Discussion sections to further highlight this point.

We note that this finding has in fact prompted us to initiate a targeted mechanistic study on the topic, which has already produced some interesting preliminary results. Besides being beyond the scope of this first manuscript, we're convinced that the addition of more context-specific follow-up experiments would only distract from the broad relevance and novelty of this first study, which presents a data-driven approach that can be applied to many relevant problems in tissue biology.